# Helminth-induced IL-4 expands bystander memory CD8+ T cells for early control of viral infection

Marion Rolot[1], Annette M. Dougall[1], Alisha Chetty[2], Justine Javaux[1], Ting Chen[1], Xue Xiao[1], Bénédicte Machiels[1], Murray E. Selkirk [3], Rick M. Maizels[4], Cornelis Hokke [5], Olivier Denis[6], Frank Brombacher[2,7,8], Alain Vanderplasschen[1], Laurent Gillet[1], William G. C. Horsnell[2,9,10] & Benjamin G. Dewals[1]

Infection with parasitic helminths can imprint the immune system to modulate bystander inflammatory processes. Bystander or virtual memory CD8+ T cells ($T_{VM}$) are non-conventional T cells displaying memory properties that can be generated through responsiveness to interleukin (IL)-4. However, it is not clear if helminth-induced type 2 immunity functionally affects the $T_{VM}$ compartment. Here, we show that helminths expand CD44$^{hi}$CD62L$^{hi}$CXCR3$^{hi}$CD49d$^{lo}$ $T_{VM}$ cells through direct IL-4 signaling in CD8+ T cells. Importantly, helminth-mediated conditioning of $T_{VM}$ cells provided enhanced control of acute respiratory infection with the murid gammaherpesvirus 4 (MuHV-4). This enhanced control of MuHV-4 infection could further be explained by an increase in antigen-specific CD8+ T cell effector responses in the lung and was directly dependent on IL-4 signaling. These results demonstrate that IL-4 during helminth infection can non-specifically condition CD8+ T cells, leading to a subsequently raised antigen-specific CD8+ T cell activation that enhances control of viral infection.

[1] Immunology-Vaccinology, Department of Infectious and Parasitic Diseases, Faculty of Veterinary Medicine – FARAH, University of Liège, Avenue de Cureghem 10, 4000 Liège, Belgium. [2] Institute of Infectious Disease and Molecular Medicine and Division of Immunology, University of Cape Town, Cape Town, South Africa 7925. [3] Department of Life Sciences, Imperial College London, London SW7 2AZ, UK. [4] Wellcome Centre for Molecular Parasitology, Institute of Infection, Immunity and Inflammation, University of Glasgow, Glasgow G12 8TA, UK. [5] Department of Parasitology, Leiden University Medical Center, Albinusdreef 2, 2333 ZA Leiden, The Netherlands. [6] Scientific Institute of Public Health, Immunology, Communicable and Infectious Diseases, Rue Engeland 642, 1180 Brussels, Belgium. [7] International Centre for Genetic Engineering and Biotechnology, 7925 Cape Town, South Africa. [8] South African Medical Research Council (SAMRC), 7925 Cape Town, South Africa. [9] Institute of Microbiology and Infection, University of Birmingham, B15 2TT Birmingham, UK. [10] Laboratory of Molecular and Experimental Immunology and Neurogenetics, UMR 7355, CNRS-University of Orleans and Le Studium Institute for Advanced Studies, Rue Dupanloup, 45000 Orléans, France. These authors contributed equally: Marion Rolot, Annette M. Dougall. Correspondence and requests for materials should be addressed to B.G.D. (email: bgdewals@uliege.be)

Soil-transmitted helminths and schistosomes infect more than a quarter of the world population, essentially afflicting people who live in areas of poverty in the developing world[1]. Heavy parasite infections cause morbidity and mortality that can occur at levels high enough to delay socio-economic development[2]. Low-burden infections with helminths while mostly asymptomatic can still have bystander effects on other diseases, especially in the case of autoimmunity and allergy[3,4], thus advocating the use of specific helminths or derived products as therapeutic strategies while encouraging guided deworming campaigns[5]. However, how bystander helminth infections modulate the control of heterologous pathogens such as viruses is understood in only a limited number of contexts and reports of both beneficial and detrimental effects on viral pathology exist[6–10].

Memory establishment and maintenance is the hallmark of the adaptive immune system and essential for ultimate control of many pathogens. B and T lymphocytes are unique in their ability to acquire immune memory against specific antigens (Ag) in order to provide these high levels of protection. However, these lymphocytes can also launch less stringent, but still effective responses to either antigen or host immune responses[11,12]. Furthermore, conditioning of T cells can impart memory-like properties and functions in absence of encounter of their cognate Ag[13], and be important for priming CD4[+] T cells for subsequent type 2 immunity[14]. This is also the case for CD8[+] T cells; bystander or virtual memory CD8[+] T cells ($T_{VM}$) emerge from early in life in naive mice[15–18] and humans[19,20] in the absence of specific Ag stimulation and are thus Ag-inexperienced. $T_{VM}$ cells have a memory-like phenotype with more effective responses to Ag encounter compared to naïve cells and characterized by expression of high levels of CD44 and also CD62L but low levels of CD49d (α4 integrin). $T_{VM}$ emerge in naive mice with an unrestricted TCR repertoire and in response to various stimuli including IL-15, IFN-I, and IL-4[13,20–22]. While TCR involvement remains to be fully deciphered, recent data suggest that $T_{VM}$ are favored by stronger TCR signals against self-antigens but maintain self-tolerance[13,21–24]. Whereas $T_{VM}$ development in C57BL/6 mice mostly depends on IL-15, IL-4 is the main driver of $T_{VM}$ expansion in BALB/c mice[25].

Parasitic helminths induce type 2 immunity characterized by high levels of IL-4[26]. Bystander consequences of this strong induction of IL-4 on memory CD8[+] T cells is not well understood in the context of helminth infection that also drive strong regulatory responses. In this study, we show that infection with helminths (*Schistosoma mansoni*, *Nippostongylus brasiliensis*, and *Heligmosomoides polygyrus*) or immunization with *S. mansoni* Ags, expands bystander $T_{VM}$ cells in secondary lymphoid tissues via IL-4. This Ag-nonspecific conditioning of CD8[+] T cells prior to encounter of their specific Ag provides early and enhanced control of a subsequent gammaherpesvirus acute infection. This enhanced protection was the result of higher levels of virus-specific CD8[+] T cell effector responses. Thus, during helminth infection IL-4 can expand and condition $T_{VM}$ cells for more rapid CD8 responses against subsequent cognate Ag encounter.

## Results

**S. mansoni eggs induce $T_{VM}$ in peripheral lymphoid tissues.** To investigate how the $T_{VM}$ cellular compartment is affected by helminth-induced inflammation, we first used a well-characterized experimental model for inducing type 2 inflammation by helminth Ags, in which eggs of the trematode parasite *S. mansoni* are injected intraperitoneally (i.p.) to 6–8-week-old female BALB/c mice before intravenous challenge (i.v.) 2 weeks later, and responses measured at d22 after the first injection

(Supplementary Figure 1a)[27]. We confirmed that *S. mansoni* eggs induced eosinophilic granulomas in the lung (Supplementary Figure 1b) and typical type 2 inflammation with high levels of soluble schistosome egg Ag (SEA)-specific IgG1 (Supplementary Figure 1c) and IL-4 production upon SEA restimulation of the dLN (Supplementary Figure 1d).

The CD8[+] T cell populations were initially compared from lung, dLN and spleen of BALB/c mice subjected to *S. mansoni* egg immunization or not and according to their expression of CD44, CD62L, and CD49d (Supplementary Figure 1b). *S. mansoni*-driven type 2 inflammation did not induce significant increase of lung or spleen cellularity while cell numbers in the dLN were increased (Fig. 1a). There was no overt response in the lung, despite increased true memory CD44[hi]CD49d[hi] T cell ($T_{TM}$) proportions (Fig. 1b–d). However, responses in dLN and spleen were significantly affected with strikingly increased numbers and proportions of CD44[hi]CD49d[lo] T cells, corresponding to the $T_{VM}$ compartment. Increased levels of eomesodermin (Eomes) were also found in $T_{VM}$ cells after *S. mansoni* egg immunization, whereas $T_{VM}$ retained low-expression levels of T-bet (Fig. 1e, f), a typical feature of $T_{VM}$ cells[22].

**IL-4 directs the expansion of $T_{VM}$ after helminth exposure.** We then investigated the implication of IL-4 responsiveness in the expansion of the $T_{VM}$ pool after *S. mansoni* egg immunization. We confirmed that treatment with IL-4 complexes (IL-4c) strikingly induced CD44[hi]CD49d[lo] $T_{VM}$ cells[28] expressing high levels of Eomes (Fig. 2a, b). IL-4 drives the expansion of $T_{VM}$ cells expressing CXCR3[29]. Thus, we further included CXCR3 surface expression in our analyses and observed that the main part of CD44[hi]CXCR3[hi] cells expressed high levels of CD62L and that population expressed low levels of CD49d, corresponding to $T_{VM}$ cells (Fig. 2c). The expansion of CXCR3[hi] $T_{VM}$ cells was further observed after *S. mansoni* egg immunization restricted to i.p. injection (Supplementary Figure 2a) or immunization with SEA or the Th2-driving schistosome egg recombinant protein omega-1 (Supplementary Figure 2b). $T_{VM}$ expansion and Eomes upregulation was also observed in mice at later time points after *S. mansoni* egg immunization (d29 and d43 after initial i.p.), suggesting that conditioning of $T_{VM}$ is long lasting (Supplementary Figure 2c). Furthermore, we also observed $T_{VM}$ expansion during other helminth-driven IL-4 dominated responses such as natural infection with *N. brasiliensis* at day 10 pi (Supplementary Figure 2d), and a significant $T_{VM}$ expansion and Eomes upregulation could also be observed by day 35 pi (Supplementary Figure 2e). In addition, natural infection with *H. polygyrus* at day 15 pi (Supplementary Figure 2f, g), and *S. mansoni* at week 7 pi (Supplementary Figure 2h) also caused increased $T_{VM}$ cell responses. These results further indicated that IL-4-dominated responses to helminth Ags can drive a long-lasting expansion of $T_{VM}$ cells in peripheral lymphoid tissue.

*S. mansoni* egg injection to *Il4ra*[−/−] BALB/c mice did not result in the expansion of $T_{VM}$ cells (Fig. 2d) and unbiased restimulation of splenocytes with phorbol myristate acetate (PMA) and ionomycin resulted in increased IFN-γ production by CD8[+] T cells that was dependent on IL-4 receptor expression (Fig. 2e). We further sought to determine whether $T_{VM}$ expansion was directly dependent upon IL-4 responsiveness of CD8[+] T cells. Mixed chimeras were generated with congenically distinct BM from WT or *Il4ra*[−/−] BALB/c mice and subjected to *S. mansoni* egg immunization (Fig. 2f) or *S. mansoni* natural infection (Supplementary Figure 2i). Similar chimerism was observed in both *S. mansoni* egg- or phosphate-buffered saline (PBS)-treated mice. As in the intact mice, the frequency of $T_{VM}$ cells and Eomes expression levels in $T_{VM}$ cells were significantly

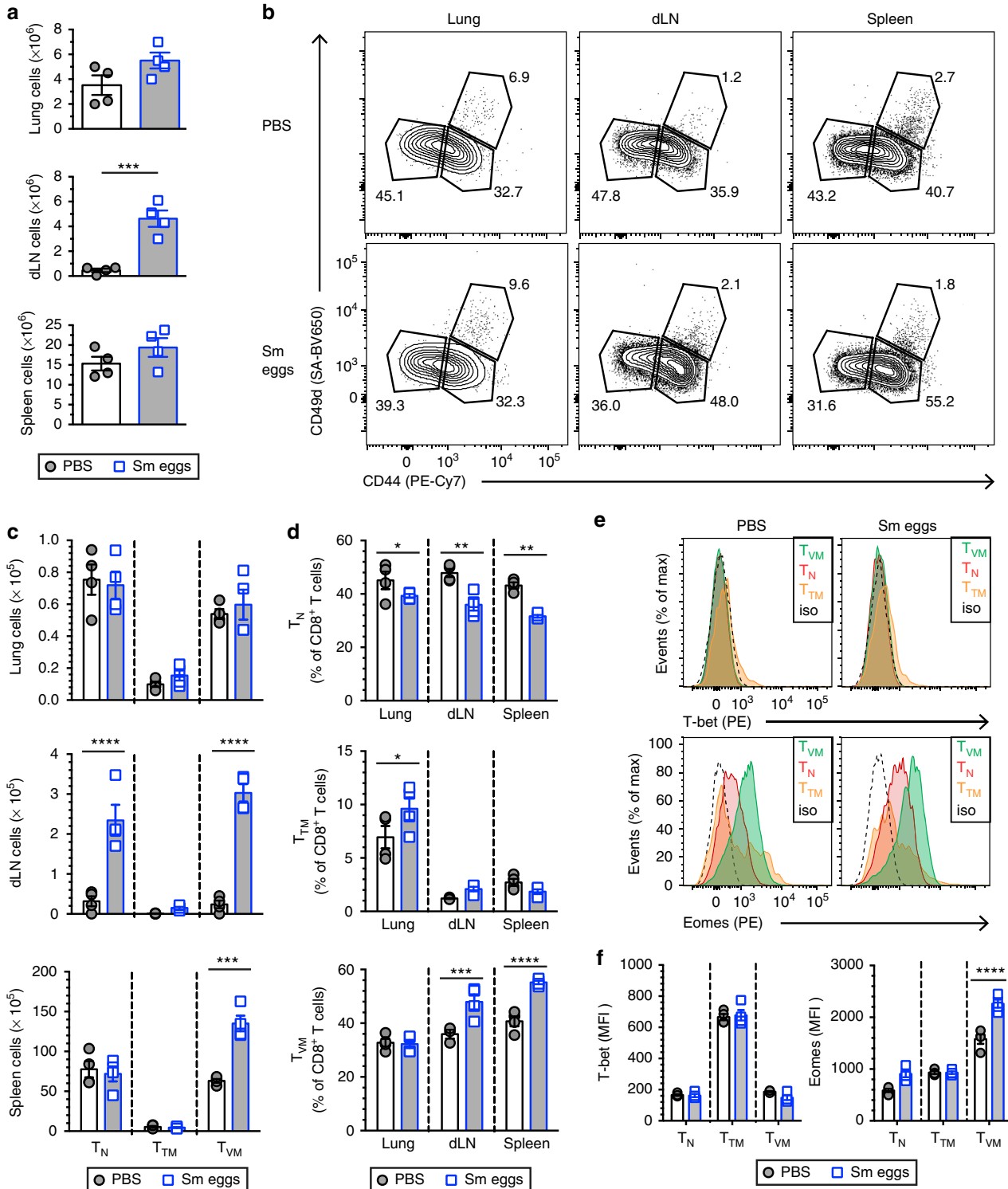

**Fig. 1** *S. mansoni* eggs induce CD44^hiCD49d^lo CD8^+ T cell expansion in the draining LN and spleen. BALB/c mice were injected with *S. mansoni* (Sm) eggs i. p. and challenged i.v. (5000 per injection) at d14 before analysis at d22. **a** Absolute cell number at d22 in the indicated tissue. **b** Representative flow contour plots of gated CD8^+ T cells of the indicated tissue at d22. Numbers indicate percent of events in each gate. **c** Cell number of CD44^lo (naive T cells, T_N), CD44^hiCD49d^hi (true memory T cells, T_TM) and CD44^hiCD49d^lo (virtual memory, T_VM) CD8^+ T cells as determined by flow cytometric analysis. **d** Percentage of CD44^lo T_N, CD44^hiCD49d^hi T_TM and CD44^hiCD49d^lo T_VM cells in CD8^+ T cells as determined by flow cytometric analysis. **e** Representative histograms of PBS- or Sm egg-treated spleen CD8^+ T cells. Gates were placed on CD44^lo T_N, CD44^hiCD49d^hi T_TM and CD44^hiCD49d^lo T_VM cells. Flow histograms show respective expression of Eomes and T-bet. **f** Median fluorescence intensities (MFI) of T-bet and Eomes at day 22 from the indicated populations after Sm egg injection. Statistical significance calculated using two-way analysis of variance (ANOVA) and Dunnett's (PBS as reference mean) or Sidak's multiple-comparison test (**P < 0.01, ***P < 0.001, ****P < 0.0001). Data are representative of three independent experiments with 4 mice per group (mean ± s.e.m. in **a**, **c**, **d**, **f**)

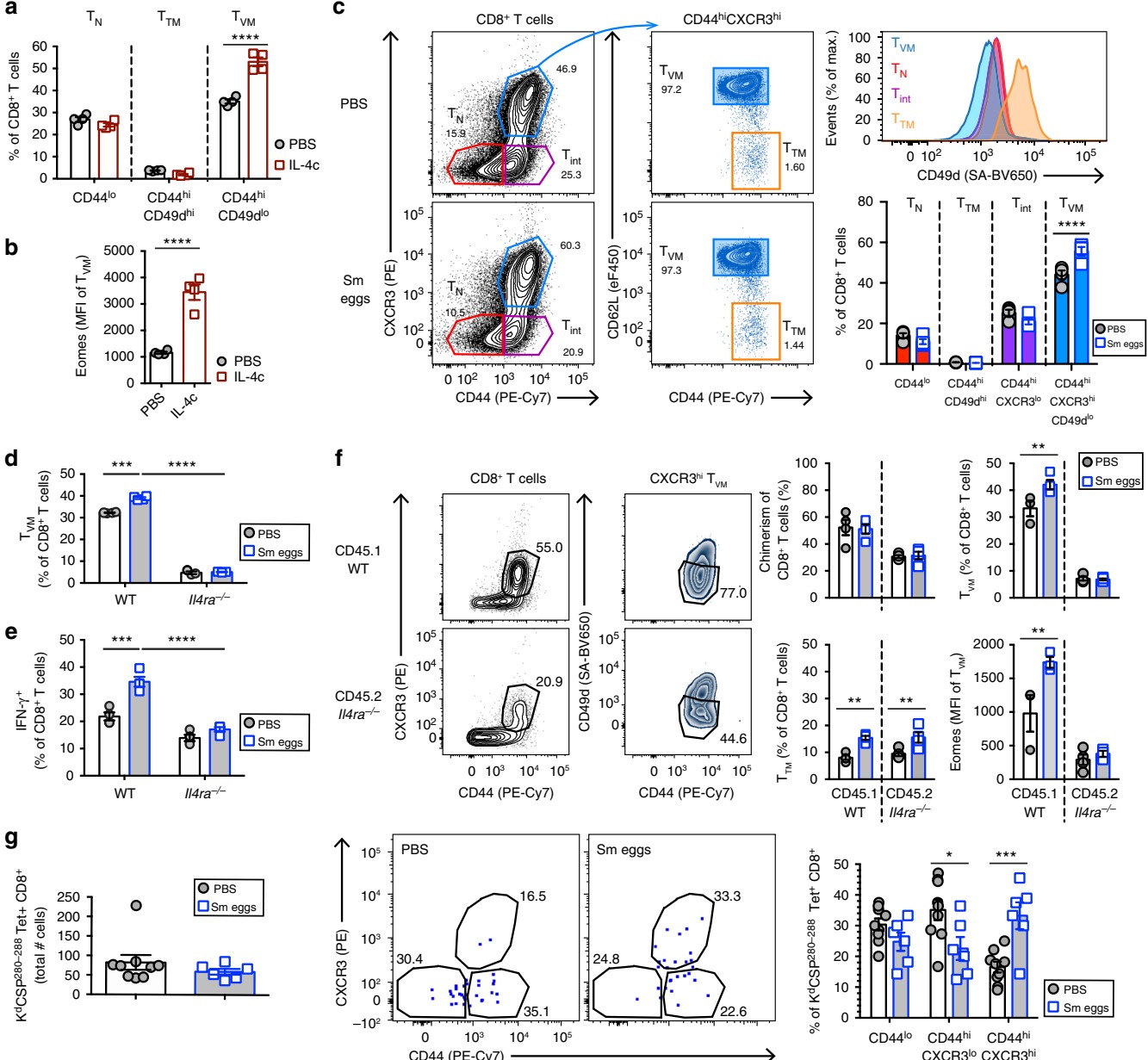

**Fig. 2** IL-4 signaling in CD8+ T cells expands $T_{VM}$ after *S. mansoni* egg immunization. **a** IL-4c treatment in BALB/c mice at d0 and d2 before analysis at d4. Percentages of spleen naive T cells ($T_N$, CD44$^{lo}$), true memory ($T_{TM}$, CD44$^{hi}$CD49d$^{hi}$) and virtual memory ($T_{VM}$, CD44$^{hi}$CD49d$^{lo}$) CD8+ T cells. **b** Intracellular Eomes in spleen $T_{VM}$ cells after IL-4c treatment as in (**a**). MFI median fluorescence intensity. **c** BALB/c mice were injected with *S. mansoni* (Sm) eggs i.p. and challenged i.v. (5000 per injection) at day 14 before analysis at day 22. Representative contour plots gated on spleen CD8+ T cells. Gates were placed depending on CD44 and CXCR3 to define CD44$^{lo}$CXCR3$^{lo}$ $T_N$, CD44$^{hi}$CXCR3$^{lo}$ $T_{int}$, CD44$^{hi}$CXCR3$^{hi}$CD62L$^{lo}$CD49d$^{hi}$ $T_{TM}$ and CD44$^{hi}$CXCR3$^{hi}$CD62L$^{hi}$CD49d$^{lo}$ $T_{VM}$ cells. Histogram overlay shows respective CD49d expression in each population. **d** Percentages of spleen CD44$^{hi}$CXCR3$^{hi}$CD62L$^{hi}$CD49d$^{lo}$ $T_{VM}$ cells in WT and *Il4ra$^{-/-}$* mice after Sm egg were injected i.p. and analysis performed at d7. **e** Percentage of IFN-γ producing splenic CD8+ T cells from WT and *Il4ra$^{-/-}$* following restimulation with PMA and ionomycin as determined by flow cytometric analysis. **f** Mixed BM chimeras were generated by introducing WT CD45.1 and *Il4ra$^{-/-}$* CD45.2 BALB/c donor BM into lethally irradiated WT CD45.1.2 BALB/c hosts. Eight weeks later, chimeric mice were injected with Sm eggs. Representative contour plots are shown from a Sm egg-treated spleen. Percentage of chimerism, CD44$^{hi}$CD49d$^{hi}$ $T_{TM}$, CXCR3$^{hi}$CD49d$^{lo}$ $T_{VM}$, and Eomes expression in $T_{VM}$ (MFI) for both donor populations in spleen are shown. **g** Tetramer-based enrichment of *P. yoelii* CSP$^{280-288}$-specific CD8+ T cells was performed on spleen and lymph nodes from individual aged-matched BALB/c mice injected or not with Sm eggs i.p. and challenged i.v. at day 14 before analysis at day 22. Total tetramer-binding cells numbers, representative flow dot plots as in Supplementary Figure 2j with numbers indicating mean percent of each gated population, and percentages of the indicated populations are shown. Statistical significance calculated using two-way analysis of variance (ANOVA) and Dunnett's (PBS as reference mean) or Sidak's multiple-comparison test (**P < 0.01, ***P < 0.001, ****P < 0.0001). Data are representative of two to three independent experiments with three to nine mice per group (mean ± s.e.m.)

increased in the WT compartment after exposure to the parasite (Fig. 2f and Supplementary Figure 2g). However, CXCR3hiCD49-dlo $T_{VM}$ cells of the $Il4ra^{-/-}$ genotype were significantly reduced in naive mice compared to WT compartment and did not expand after helminth exposure. These results demonstrated that IL-4 responsiveness of CD8+ T cells conditions and expands $T_{VM}$ cells after S. mansoni egg immunization or natural infection.

Next, in order to examine whether IL-4-dependent $T_{VM}$ expansion would not also result from an Ag-specific response to S. mansoni eggs, a tetramer-based enrichment was performed on an unrelated and randomly chosen population of CD8+ T cells expressing a TCR specific to the H-2Kd-restricted SYVPSAEQI peptide of the circumsporozoite protein (CSP280–288) of Plamodium yoelii (Supplementary Figure 2j). We observed that exposure to S. mansoni eggs also caused increased proportions of CD44hiCXCR3hi CSP280–288-specific CD8+ T cells (Fig. 2g). These results further support that expansion of $T_{VM}$ induced by helminth exposure is Ag-nonspecific.

**Helminths ameliorate the control of MuHV-4 infection.** Ag-inexperienced $T_{VM}$ cells respond more quickly to their cognate Ag than naive T cells[30] and IL-4 signaling in memory CD8+ T cells were previously suggested to reduce effector responses[28]. Thus, we investigated whether expanding the $T_{VM}$ pool through helminth exposure would affect effector CD8+ T cell responses against heterologous Ags.

Murid herpesvirus 4 (MuHV-4) is a gammaherpesvirus that infects the laboratory mouse and establishes long term persistence[31]. Interestingly, levels of primary MuHV-4 lytic infection are directly dependent on effective development of effector CD8+ T cells[32]. MuHV-4 was therefore used to assess virus-specific CD8+ T cell responses after exposure to helminth Ags. We first immunized 8-week-old female BALB/c mice with S. mansoni eggs before their infection with $1 \times 10^4$ plaque-forming units (PFU) of MuHV-4 intranasally under general anesthesia (Fig. 3a). A control group consisted of PBS-treated mice. Immunization with S. mansoni eggs protected against transient weight loss caused by MuHV-4 respiratory infection (Fig. 3b) and was associated with reduced levels of infection at day 7 postviral infection (pvi) as determined by immunostaining on lung tissue (Fig. 3c) and plaque assay (Fig. 3e). We next measured the levels of infection over time using MuHV-4-luc recombinant virus for live imaging of light emission centered on the thorax at days 2, 5, 7, and 9 pvi[33]. S. mansoni egg treatment resulted in significantly reduced levels of light emission by d7 pvi (Fig. 3d). There was a similar early control of respiratory MuHV-4 infection when BALB/c mice were infected 49 days after percutaneous exposure to S. mansoni cercariae, a time point corresponding to the peak of the response against the parasite eggs (Supplementary Figure 3a), whereas enhanced control of viral infection was not apparent when mice were coinfected at earlier time points of the parasite life cycle. Likewise, enhanced control of viral infection was also observed when mice were infected with MuHV-4 at d6 or d35 after infection with N. brasiliensis (Supplementary Figure 3b and c).

Colonization of the host by MuHV-4 after lytic respiratory infection was not significantly affected by prior exposure to S. mansoni eggs as attested by live imaging of light emission centered on the superficial cervical LN (scLN, Fig. 3f) and qPCR for viral genome detection in the spleen at days 5, 7, 11, and 20 pvi (Fig. 3g). These observations further supported a role of CD8-dependent viral control during lytic replication as effective MuHV-4-specific CD8+ T cell responses were shown to be unable to control the establishment of viral latency[34]. We further observed that increasing MuHV-4 infectious dose resulted in enhanced control (Fig. 3h) and daily imaging of mice infected

with $1 \times 10^4$ PFU showed similar infection levels up to days 5–6 pvi before being controlled in S. mansoni-exposed mice (Fig. 3i), suggesting enhanced adaptive immune responses rather than impaired viral growth.

**Helminths augment lung CD8+ T cell responses to MuHV-4.** We next assessed the immune response against MuHV-4 in BALB/c mice that were exposed to S. mansoni eggs or not. We observed no significant difference in the antibody responses against MuHV-4 (Supplementary Figure 4a) and global cellularity in lungs at day 7 pvi was not affected (Supplementary Figure 4b). Whereas eosinophils and DC numbers were significantly increased in mice exposed to S. mansoni eggs, numbers of neutrophils, macrophages, monocytes, B or CD4+ T cells were not affected (Supplementary Figure 4b). Strikingly, the frequency and number of lung CD8+ T cells was significantly increased at day 7 pvi in mice immunized with S. mansoni eggs (Fig. 4a–c) and mice infected percutaneously with S. mansoni cercariae 7 weeks before MuHV-4 infection (Fig. 4d). Such enhanced CD8+ T cell response was associated with increased proportions of CD44hiCD62Llo effector T cells (Fig. 4e). We further observed that effector CD44hiCD62LloCD49dhi CD8+ T cell responses were transiently but significantly increased by day 7 pvi in the bronchoalveolar lavage fluid (BALF) and lungs of mice prior exposed to S. mansoni eggs (Fig. 4f), as well as short-lived effector T cells (KLRG1+CD127−) (Fig. 4g). These results suggest that prior exposure to helminths enhances CD8+ T cell responses after MuHV-4 infection.

**Enhanced virus-specific CD8 response after helminth exposure.** We further sought to evaluate the effector role of CD8+ T cell responses against MuHV-4 infection in BALB/c mice after exposure to S. mansoni eggs. Interferon (IFN)-γ and granzyme B (GzmB) expression levels in BAL were significantly increased by day 7 pvi (Fig. 5a) and unbiased restimulation of lung cells with PMA and ionomycin caused significantly increased co-production of IFN-γ and TNF-α by CD8+ T cells (Supplementary Figure 5a, b).

In order to assess the MuHV-4-specific CD8+ T cell response, we took advantage of H-2b congenic BALB/B mice in which the response against the well-established MuHV-4 immunodominant H-2Db-restricted AGPHNDMEI (ORF6487–495) and H-2Kb-restricted SVYGFTGV (ORF61524–531) epitopes could be measured[35]. We initially measured thoracic light emission after MuHV-4-luc infection of BALB/B mice and observed similar enhanced control of viral infection at day 7 pvi (Fig. 5b). Strikingly, tetramer staining (Fig. 5c, d) and peptide restimulation (Fig. 5e, f) revealed significantly increased MuHV-4-specific responses in the BALF and lung in BALB/B mice that were initially exposed to S. mansoni eggs. Similar increased MuHV-4-specific responses by day 7 pvi were observed when mice where infected at day 29 or 43 after the initial i.p. injection of S. mansoni eggs (Supplementary Figure 5c). Moreover, infection of BALB/B mice with $1 \times 10^4$ PFU of MuHV-4 intranasally after their initial infection with N. brasiliensis resulted in higher MuHV-4-specific CD8+ T cell responses at day 7 pvi (Supplementary Figure 5d–h).

The results in Fig. 2e showed that S. mansoni eggs induced expansion of $T_{VM}$ cells expressing a TCR able to recognize MHC tetramers presenting the P. yoelii CSP280–288 peptide, absent from S. mansoni Ags. In order to examine the development of effector Ag-specific responses from CSP280–288-specific CD8+ T cells, we infected BALB/c mice with a MuHV-4-luc-CSP virus expressing a modified luciferase protein in which the H-2Kd-restricted CSP280–288 peptide was inserted in-frame (Supplementary Figure 5i). We observed enhanced control of MuHV-4-luc-CSP infection in the

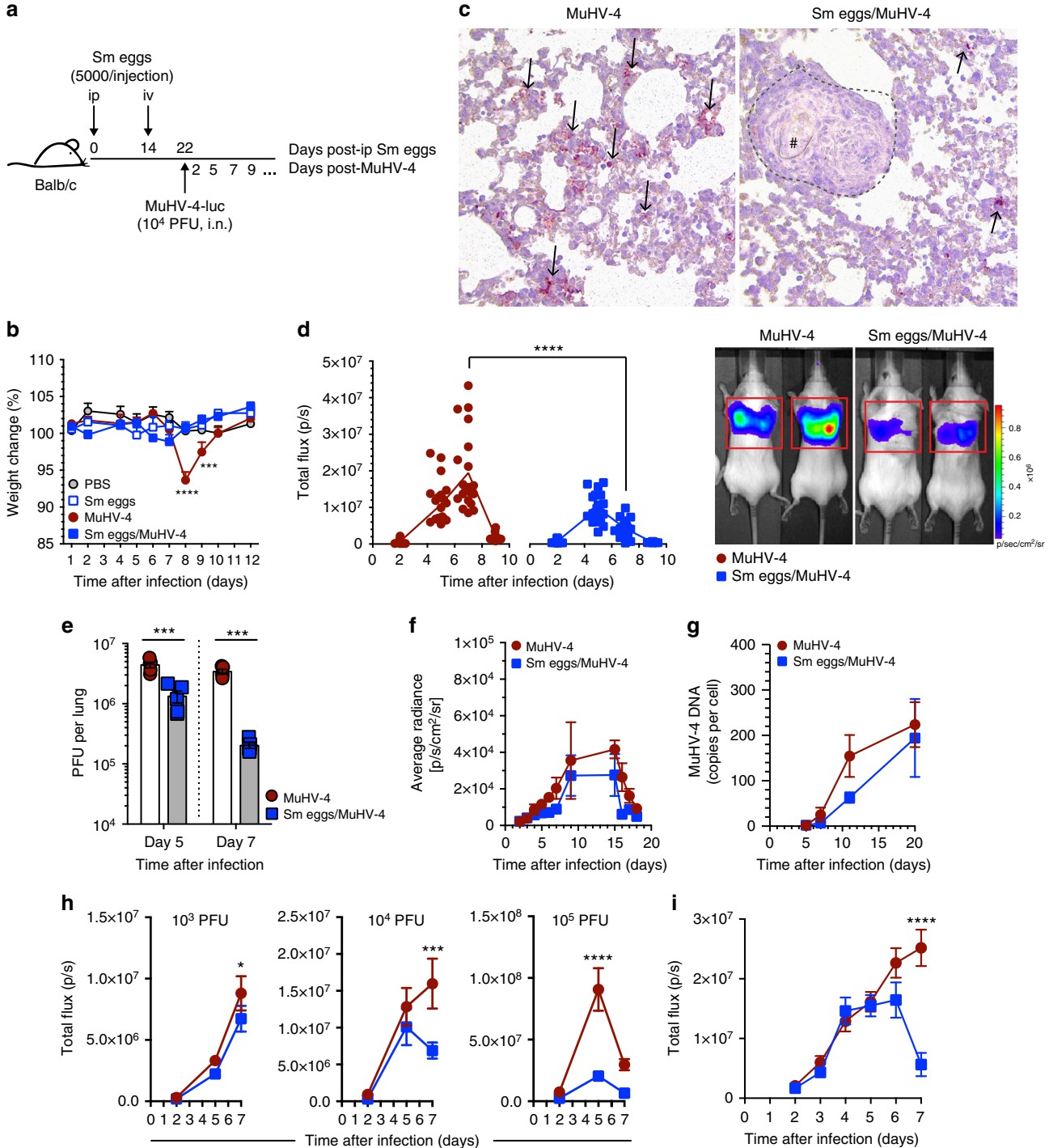

**Fig. 3** *S. mansoni* egg immunization ameliorates the control of MuHV-4 lung infection. BALB/c mice were injected with *S. mansoni* (Sm) eggs i.p. and challenged i.v. (5,000 per injection) at day 14. At d22, MuHV-4-luc virus was administered i.n. (**b–g, i** $10^4$ PFU per mouse; **h** $10^3$, $10^4$, or $10^5$ PFU per mouse in 30 μL PBS). **a** Experimental design. **b** Body weight change as percentage of initial weight. **c** Immunostaining of MuHV-4-infected cells in representative lung sections of PBS- or Sm egg-treated mouse 7 days after MuHV-4-luc infection (day 7 pvi). Arrows indicate positive AEC signal. Dotted line shows boundaries of an egg-centered (#) granuloma. **d** Combined dorsal and ventral measurements by live imaging of thoracic light emission following D-luciferin injection. p s$^{-1}$ = photons per second. Representative photographs of bioluminescence signals of two mice per group are shown. **e** Lung titers at day 7 pvi as determined by plaque assay. **f** Live imaging of light emission of superficial cervical LN (scLN). **g** Splenic MuHV-4 DNA copy numbers at the indicated time points. **h** Live imaging of thoracic light emission following MuHV-4-luc infection with three different infectious doses. **i** Live imaging of thoracic light emission of lungs following MuHV-4-luc infection daily. Statistical significance calculated using two-way analysis of variance (ANOVA) and Sidak's multiple-comparison test ($^{**}P < 0.01$, $^{***}P < 0.001$, $^{****}P < 0.0001$). Data are representative of three independent experiments with five to ten mice per group (mean ± s.e.m. in **b, e–i**)

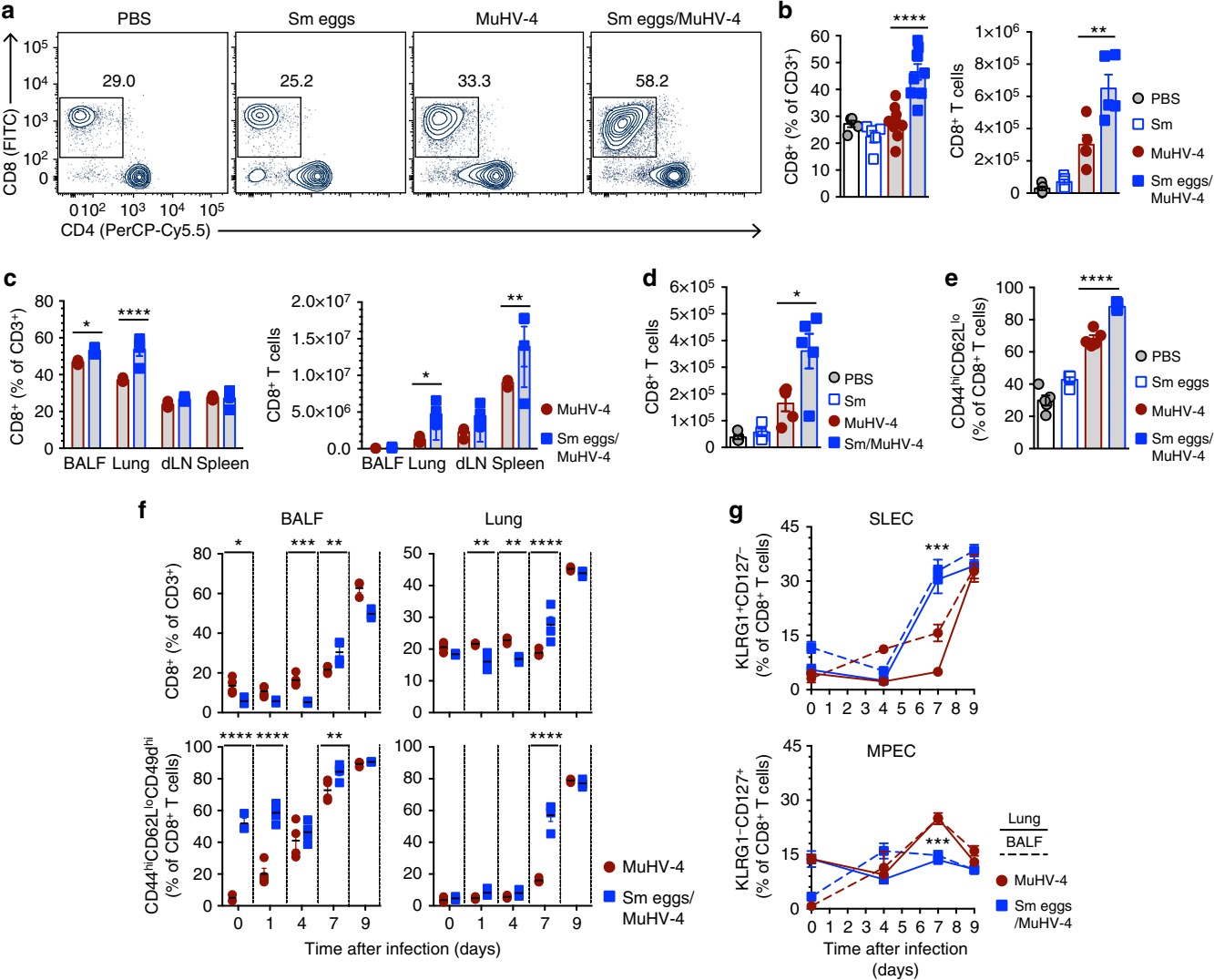

**Fig. 4** *S. mansoni* egg immunization augments effector/memory CD8+ T cell responses to MuHV-4 infection. BALB/c mice were treated with *S. mansoni* (Sm) eggs as outlined in Fig. 3a (**a–c**, **e–g**) or infected with *S. mansoni* cercariae (35 cercariae per mouse, percutaneous infection; **d**) before being infected with MuHV-4-luc virus. **a** Representative flow contour plots of gated lung CD3+ T cells at day 7 pvi. Gated population indicates the percentage of CD8+ T cells. **b** Percentage and numbers of lung CD8+ T cells at day 7 pvi as determined by flow cytometric analysis. **c** Percentage and numbers of CD8+ T cells at day 7 pvi in the indicated organs as determined by flow cytometric analysis. **d** Number of lung CD8+ T cells at day 7 pvi in mice infected with *S. mansoni* 7 weeks before MuHV-4 infection. **e** Percentage of lung effector/memory CD8+ T cells (CD44hiCD62Llo) as determined by flow cytometric analysis. **f** Percentage of BALF and lung CD8+ T cells at the indicated time points after MuHV-4 infection. **g** Percentage of short-lived (SLEC, KLRG1+CD127−) and memory precursor (MPEC, KLRG1−CD127+) effector CD8+ T cells after MuHV-4 infection. Statistical significance calculated using two-way analysis of variance (ANOVA) and Sidak's multiple-comparison test (**$P < 0.01$, ***$P < 0.001$, ****$P < 0.0001$). Data are representative of two to three independent experiments with four to five mice per group (mean ± s.e.m. in **b–g**)

lung by day 7 pvi which was associated with an increased response of effector CSP[280–288]-specific CD8+ T responses (Fig. 5g), further supporting that T_{VM} are conditioned by exposure to helminths, which could explain the enhanced effector responses against their cognate Ag.

To further investigate whether the enhanced Ag-specific CD8+ T cell response against MuHV-4 was responsible for the early viral control observed in mice treated with *S. mansoni* eggs, mice were treated with depleting antibodies against CD8 or CD4 one day before MuHV-4 infection and then at days 1 and 4 pvi (Fig. 5h). While depletion of CD4 did not inhibit the early control of MuHV-4 infection observed in mice exposed to *S. mansoni* eggs, the absence of CD8+ cells resulted in the total loss of such early control. These results demonstrate a CD8-dependent

mechanism of enhanced control of viral infection and strongly suggest that *S. mansoni* egg immunization rendered CD8+ T cell responses more efficient to clear MuHV-4 infection, independently of CD4+ T cells.

**Helminth exposure affects the gene-expression profile of T_{VM}.**
To further investigate T_{VM} phenotypic changes after exposure to *S. mansoni* eggs, we compared gene-expression profiles between PBS- or *S. mansoni* egg-treated T_{VM} cells by RNA sequencing of FACsorted T_{VM} cells from the spleen (Supplementary Figure 6). In total, we observed 29 differentially expressed (DE) genes (log₂-fold change > ± 0.5, $P < 0.1$) (Fig. 6a, b) and principal-component analysis revealed clustering of PBS- or *S. mansoni* egg-treated T_{VM} cells (Fig. 6c). Among genes upregulated in *S. mansoni*

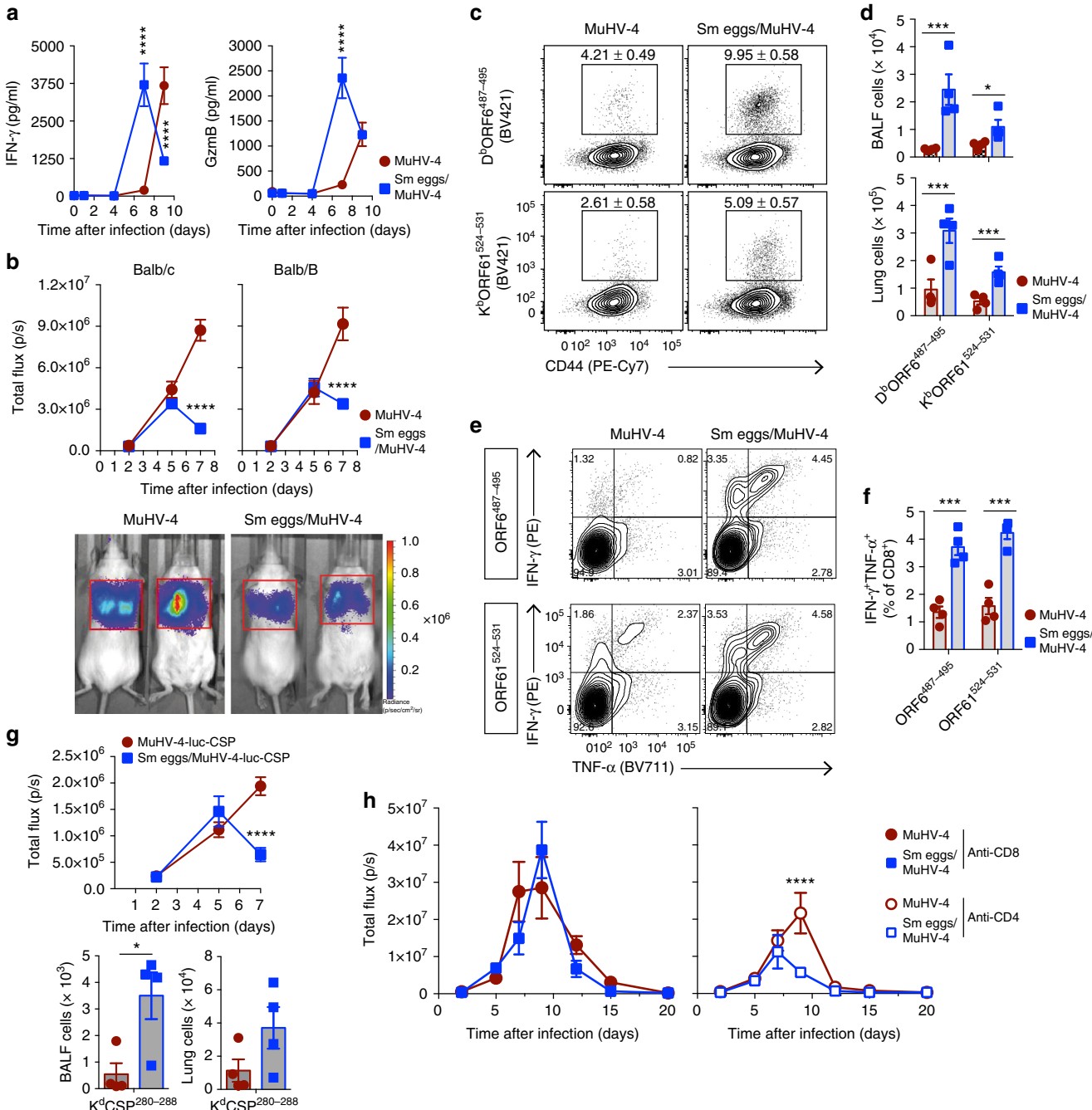

**Fig. 5** *S. mansoni* immunization augments specific antiviral CD8+ T cell responses to control lytic MuHV-4 infection. BALB/c or BALB/B mice were treated with *S. mansoni* (Sm) eggs before being infected with MuHV-4-luc or MuHV-4-luc-CSP virus as outlined in Fig. 3a. **a** IFN-γ and GzmB concentrations in the BALF after MuHV-4 infection determined by ELISA. **b** Live imaging of thoracic light emission after MuHV-4-luc infection of BALB/c or BALB/B mice prior exposed to Sm eggs. Representative photographs of bioluminescence signals of two mice per group at day 7 pvi are shown. **c** Representative flow contour plots of D$^b$ORF6$^{487-495}$ and K$^b$ORF61$^{524-531}$ MuHV-4-specific tetramer stainings of lung cells at day 7 pvi. Numbers in gate indicate percentage of tetramer-positive in CD8+ T cells. Mean ± s.e.m. are shown. **d** Number of MuHV-4-specific CD8+ T cells in the BALF and lung at day 7 pvi in BALB/B mice based on flow cytometric analysis in panel **c**. **e** Representative flow contour plots of gated lung CD8+ T cells at day 7 pvi showing IFN-γ and TNF-α intracellular stainings after ORF6$^{487-495}$ or ORF61$^{524-531}$ peptide restimulation. Numbers indicate percentage in each quadrant. **f** Percentage of IFN-γ and TNF-α co-producing lung CD8+ T cells after ORF6$^{487-495}$ or ORF61$^{524-531}$ peptide restimulation and intracellular staining and flow cytometric analysis as detailed in **e**. **g** Live imaging of thoracic light emission after MuHV-4-luc-CSP infection of BALB/c mice and number of CSP$^{280-288}$-specific CD8+ T cells in the BALF and lung at day 7 pvi. **h** Anti-CD8 (YTS-169.4) or anti-CD4 (GK1.5) antibodies were injected at day -1, days 1 and 4 after MuHV-4 infection and live imaging performed for thoracic light emission. Statistical significance calculated using two-way analysis of variance (ANOVA) and Sidak's multiple-comparison test ($^*P < 0.05$, $^{**}P < 0.01$, $^{***}P < 0.001$, $^{****}P < 0.0001$). Data are representative of two independent experiments with four to five mice per group (mean ± s.e.m. in **a, b, d–h**)

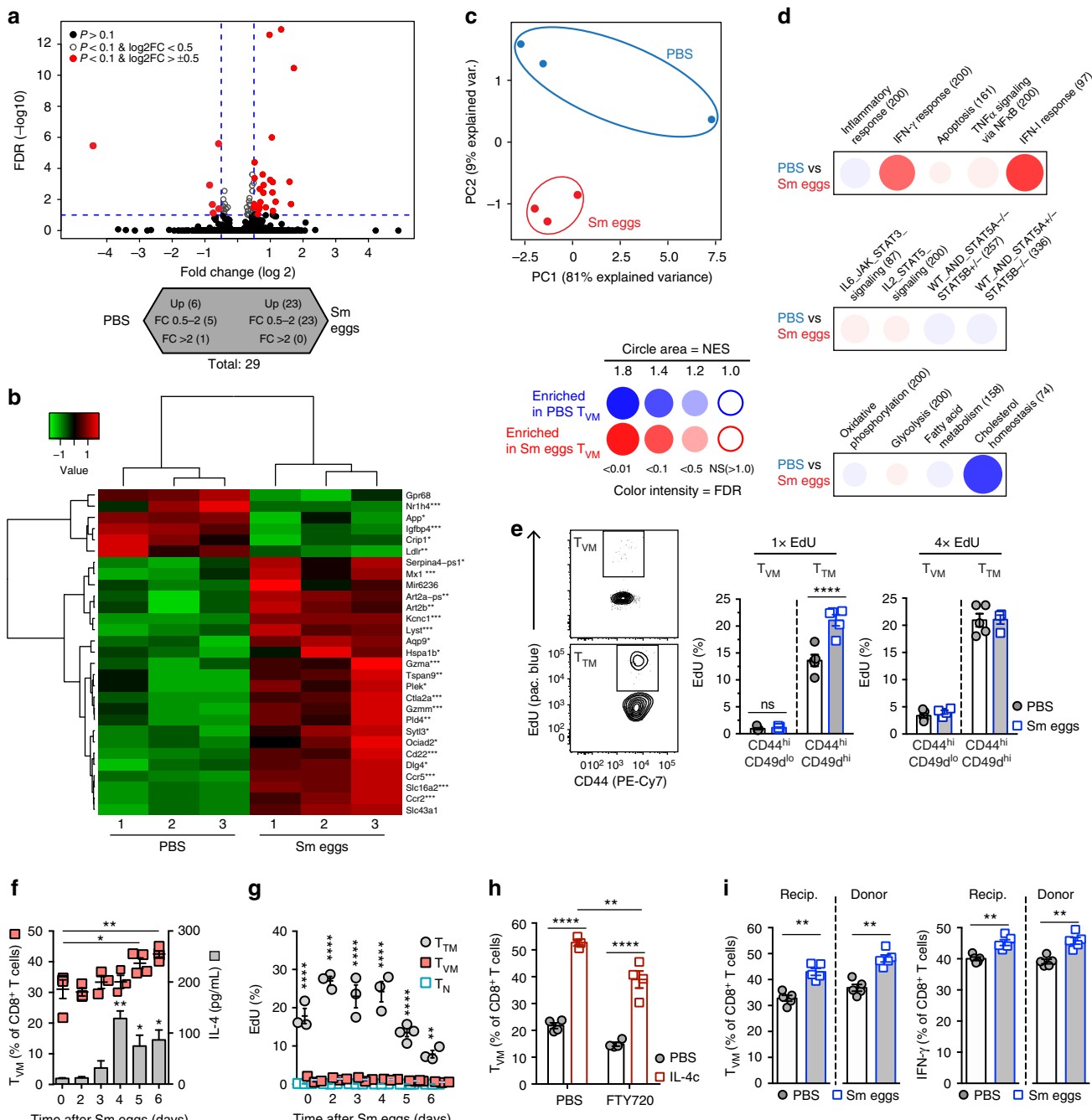

**Fig. 6** Phenotypic changes in helminth-driven $T_{VM}$ cells. **a–d** Transcriptomics analysis of FACsorted spleen $T_{VM}$ treated as in Supplementary Figure 1a. **a** Differentially expressed (DE) genes ($P < 0.1$) in red in volcano plot, and showing in the arrowheads the direction of upregulated expression. **b** Heatmap of DE genes ($P < 0.1$, DE over 1.5-fold). Left margin shows hierarchical clustering and right annotation indicate gene symbols. **c** Principal-component analysis. **d** Enrichment for selected hallmark gene-sets from the MSigDB by gene-set enrichment analysis with BubbleGUM. WT_AND_STAT5A$^{-/-}$_STAT5B$^{+/-}$ and WT_AND_STAT5A$^{-/-}$_STAT5B$^{+/-}$ gene-sets were obtained from published data[60]. Key color indicates cell subset showing enrichment for the gene set, and size of symbols and color intensity indicate significance of enrichment (surface area proportional to absolute value of the normalized enrichment score (NES); color intensity indicates the false-discovery rate (FDR)). Numbers in parentheses (above) indicate number of genes. NS not significant. **e** EdU incorporation in CXCR3$^{hi}$CD49d$^{lo}$ $T_{VM}$ and CD49d$^{hi}$ $T_{TM}$ cells after Sm egg treatment as outlined in Supplementary Figure 1a. 1 × EdU, injection 4 h before harvest; 4 × EdU, injection at days 18, 19, 20, and 21 after Sm egg i.p. injection. **f** Percentages of spleen $T_{VM}$ cells after Sm egg injection i.p. and IL-4 levels after SEA restimulation of splenocytes at the indicated time points. **g** Percentage of spleen EdU$^+$ $T_{TM}$, $T_{VM}$, or $T_N$ cells at the indicated time points after Sm egg injection i.p. **h** Percentages of spleen $T_{VM}$ cells after IL-4c treatment in mice treated daily with FTY720. **i** Spleen CD8$^+$ T cells were adoptively transferred to congenic mice before injection of Sm eggs i.p. At 7 days, percentages of recipient or donor spleen $T_{VM}$ cells and IFN-γ-producing cells upon restimulation in the spleen are shown in CD45.1.2$^+$ recipient or CD45.1$^+$ donor CD8$^+$ T cells. Statistical significance calculated using two-way analysis of variance (ANOVA) and Sidak's multiple-comparison test ($^*P < 0.05$, $^{**}P < 0.01$, $^{***}P < 0.001$, $^{****}P < 0.0001$). Data are representative of two independent experiments with three to five mice per group (mean ± s.e.m. in **e, f** and results from individual mouse are shown)

egg-induced $T_{VM}$ cells, we observed some genes related to cytotoxicity functions (*Gzma*, *Ctla2a*, and *Slc16a2*), cytokine–cytokine receptor interaction (*Ccr2* and *Ccr5*) or IFN-I responses (*Mx1*) (Fig. 6b). Further analysis of molecular signatures with BubbleGUM[36], a tool allowing gene-set enrichment analysis of transcriptomic data, revealed that among the selected gene-sets exposure to *S. mansoni* eggs induced a shift toward higher expression of genes implicated in IFN-γ and IFN-I responses, whereas reduced expression of genes involved in cholesterol homeostasis (Fig. 6d). These data demonstrate that naive or *S. mansoni* egg-induced $T_{VM}$ cells are phenotypically distinct and further suggest that this memory T cell population has enhanced capacity to initiate an antiviral response. Besides affecting their transcriptomics phenotype, exposure to Sm eggs leads to $T_{VM}$ expansion that could be due to IL-4 provoking their proliferation. However, EdU incorporation in vivo after *S. mansoni* egg immunization did not show overt proliferation in $T_{VM}$ cells, but did so in $T_{TM}$ cells after a single EdU administration 4 h before harvest (Fig. 6e). After a single injection of *S. mansoni* eggs, we observed that levels of IL-4 response to SEA increased by day 4 after injection associated with the expansion of $T_{VM}$ cells (Fig. 6f). However, only $T_{TM}$ cells showed significant proliferation and EdU incorporation over time after *S. mansoni* egg injection remained low in $T_N$ and $T_{VM}$ (Fig. 6g). $T_{VM}$ expansion could also result from a recruitment to the secondary lymphoid organs. To address this hypothesis, we treated mice with the sphingosine-1 phosphate receptor signaling FTY720 to inhibit recruitment of lymphocytes to the spleen. FTY720 treatment did not impair the IL-4-dependent expansion of $T_{VM}$ in the spleen (Fig. 6h), although the extent of the expansion was slightly affected by inhibition of lymphocyte trafficking. In addition, bulk CD8$^+$ T lymphocytes isolated from the spleen of naive mice were transferred to congenic naive BALB/c mice before immunization with *S. mansoni* eggs. $T_{VM}$ expansion occurred irrespective of their donor or recipient origin (Fig. 6i), further suggesting that helminth-induced $T_{VM}$ expansion can occur in peripheral CD8$^+$ T lymphocytes. Thus, these results suggest that the observed $T_{VM}$ expansion likely result from the conversion from naive T cells rather than proliferation or recruitment of $T_{VM}$ cells.

**IL-4 signals in CD8$^+$ T cells for early control of MuHV-4.** Next, we investigated the role of IL-4 responsiveness in the early control mediated by CD8$^+$ T cells after exposure to helminth Ags. We first used *S. mansoni* eggs to immunize WT or *Il4ra*$^{-/-}$ BALB/c mice before subjecting them to intranasal MuHV-4 infection (Fig. 7a). *Il4ra*$^{-/-}$ mice did not display the enhanced control of MuHV-4 infection observed in WT mice, with *Il4ra*$^{-/-}$ mice exposed to *S. mansoni* eggs displaying similar to higher thoracic light signals. The lack of helminth-mediated early control of MuHV-4 infection in *Il4ra*$^{-/-}$ mice was further associated with the absence of increased CD8$^+$ T cells in the lung at day 7 pvi (Fig. 7b). These results suggest that IL-4 signaling during helminth-elicited inflammation governs the early capacity of BALB/c mice to generate effector CD8$^+$ T cells and control viral infection.

To examine whether the presence of *S. mansoni* eggs in the pulmonary niche conditions the early control of MuHV-4 infection after exposure to helminth Ag, we injected *S. mansoni* eggs to 8-week-old female BALB/c mice twice via the i.p. route at 2 weeks interval before MuHV-4-luc infection intranasally at day 22 (Supplementary Figure 7a). We found that the presence of *S. mansoni* eggs in the lung was dispensable for the early control of viral infection, with significantly reduced light emission signals by day 7 pvi. In addition, treatment with IL-4c at 2 days interval before MuHV-4-luc infection intranasally at day 4 reduced the

levels of light emission reporting infection (Fig. 7c), which was associated with enhanced effector CD8$^+$ T cell responses in the lung after MuHV-4 infection (Fig. 7d, e). These results further suggest the role of IL-4-induced $T_{VM}$ cells in contributing to the early effector CD8$^+$ T cell responses against MuHV-4 infection.

The impact of IL-4 sensitivity of CD8$^+$ T cells after immunization with *S. mansoni* eggs on subsequent responses against MuHV-4 infection was further investigated by co-transfer of $T_{VM}$-rich WT and $T_{VM}$-poor *Il4ra*$^{-/-}$ purified CD8$^+$ T cells from PBS− or *S. mansoni* egg-treated mice cells in equivalent numbers to congenic PBS− or *S. mansoni* egg-treated BALB/c mice (Supplementary Figure 7b). Mice were then infected or not with MuHV-4. The absolute numbers of donor WT and *Il4ra*$^{-/-}$ CD8$^+$ T cells localizing to the lung tended to increase at 7 day pvi (Fig. 7f), and analysis of WT:*Il4ra*$^{-/-}$ donor cell ratios normalized to uninfected mice demonstrated significant enrichment of donor WT CD8$^+$ T cells compared to donor *Il4ra*$^{-/-}$ CD8$^+$ T cells when the mice were initially treated with *S. mansoni* eggs (Fig. 7g). Importantly, IL-4 signaling in CD8$^+$ T cells resulted in significantly increased IFN-γ production after unbiased restimulation of lung cells from *S. mansoni* egg-treated mice after MuHV-4 infection, compared to the donor *Il4ra*$^{-/-}$ CD8$^+$ T cell compartment (Fig. 7h). These results demonstrated that IL-4 signaling in CD8$^+$ T cells, probably through expansion of $T_{VM}$ cells, contributes significantly to condition effective antiviral CD8$^+$ T cell responses and that IL-4Rα-dependent *S. mansoni* egg-induced inflammation in the lung environment alone is not sufficient to significantly enhance CD8$^+$ T cell responses. Nonetheless, it was unclear whether *S. mansoni* egg-induced $T_{VM}$ cells could outcompete their naive counterparts. Thus, we adoptively co-transferred $T_{VM}$ cells FACsorted from PBS− and *S. mansoni* egg-treated mice cells in equivalent numbers to congenic naive BALB/c mice. At day 7 after MuHV-4 infection, effector T cells originating from transferred $T_{VM}$ cells could be detected in lungs and BALF demonstrating the contribution of $T_{VM}$ cells to the effector antiviral CD8 response (Fig. 7i). However, no significant differences could be observed between naive or Sm egg-induced $T_{VM}$ compartments (Fig. 7j). Thus, although phenotypically distinct to naive $T_{VM}$ cells, preliminary expansion of $T_{VM}$ cells contributes to the early control of MuHV-4 infection.

## Discussion

IL-4 and IFN-γ are usually considered antagonistic as hallmark cytokines of type 2 and type 1 immunity, respectively. Nonetheless, IL-4 can drive Eomes expression in CD8$^+$ T cells and lead to IFN-γ production[23,37]. We confirm here that this intriguing mechanism has functional consequences in vivo by providing evidence that IL-4 induced by helminths can have under certain circumstances, beneficial bystander consequences on IFN-γ-dependent antiviral effector responses through induction of $T_{VM}$ expansion. Immunity against helminths could therefore have evolved a safety mechanism through induction of highly responding $T_{VM}$ cells to counterbalance negative effects of type 2 immunity on the development of effective antiviral responses.

Helminth infections are highly prevalent and have been shown to modulate the immune system, sometimes leaving a long-lasting imprint on the ability of the helminth-exposed hosts to respond to heterologous Ags. Indeed, helminth infections can down-modulate allergy or inflammatory bowel disease through various mechanisms[38], but have also been involved in modulating the ability of the infected host to control virus infections. Several groups have shown using C57BL/6 mice that exposure to *H. polygyrus* or *S. mansoni* eggs enhanced reactivation from latency of MuHV-4 through changes in the IL-4/IFN-γ balance[10] and also that intestine dwelling helminths could alter effector CD8$^+$ T

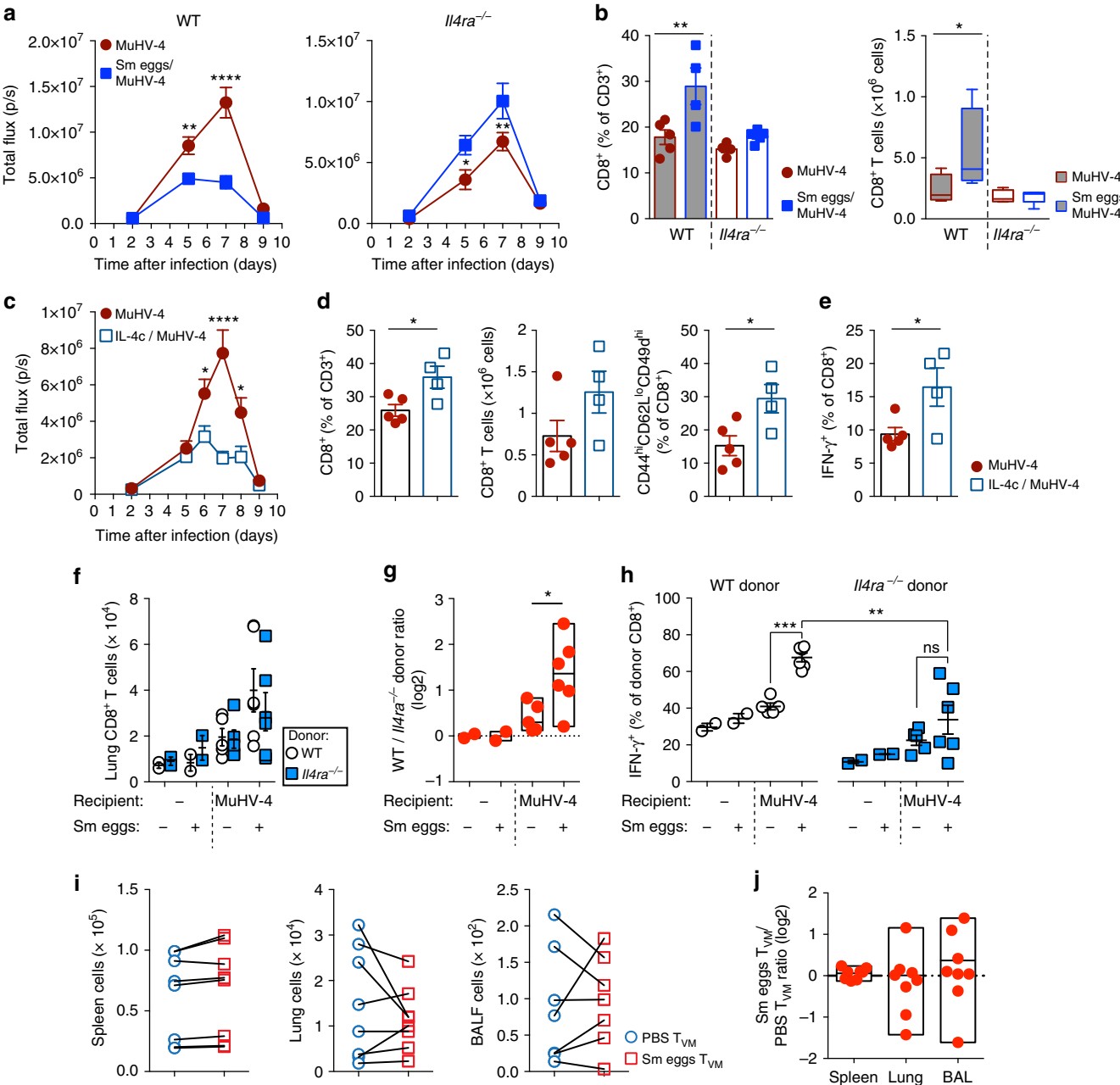

**Fig. 7** Helminth-induced IL-4 conditions CD8$^+$ T cells for early control of MuHV-4 infection. BALB/c mice were treated with *S. mansoni* eggs (Sm) (**a**, **b**) or treated with IL-4c (**c**–**e**) before being infected with MuHV-4-luc virus. **a** Live imaging of combined dorsal and ventral thoracic light emission of WT or *Il4ra*$^{-/-}$ mice after MuHV-4-luc infection. **b** Lung CD8$^+$ T cell responses at day 7 pvi in WT or *Il4ra*$^{-/-}$ mice. **c** Live imaging of combined dorsal and ventral thoracic light emission of mice subjected to the outlined treatments. **d** Lung CD8$^+$ T cell responses and proportions of CD44$^{hi}$CD62L$^{lo}$CD49d$^{hi}$ effector CD8$^+$ T cells at day 7 pvi in naive or IL-4c treated mice as in **c**. **e** Percentage of IFN-γ producing lung CD8$^+$ T cells by intracellular staining after restimulation with PMA and ionomycin at day 7 pvi. **f**–**h** Spleen CD8$^+$ T cells were purified from WT CD45.1$^+$ or *Il4ra*$^{-/-}$ CD45.2$^+$ mice treated with PBS or Sm eggs before co-transfer in PBS– or Sm egg-treated congenic CD45.1.2 WT recipient mice, followed by MuHV-4 infection as outlined in Supplementary Figure 7b. **f** Donor cell numbers in lungs at 7 day pvi. **g** Ratio of donor cells in lungs at 7 day pvi normalized on uninfected control group (PBS or Sm egg-treated uninfected mice for PBS or Sm egg-treated MuHV-4 infected mice). **h** Percentage of IFN-γ producing donor CD8$^+$ T cells following restimulation with PMA and ionomycin. **i**, **j** CD44$^{hi}$CXCR3$^{hi}$CD62L$^{hi}$CD49d$^{lo}$ T$_{VM}$ cells were FACS-sorted from PBS-treated BALB/c CD45.1$^+$ and Sm egg-treated CD45.1.2$^+$ mice, and equal numbers of cells were co-transferred into naive CD45.2$^+$ recipients. One day after adoptive transfer, recipients were infected with MuHV-4. **i** Total numbers of donor cells in spleen, lung and BALF at day 7 pvi. **j** Sm egg-T$_{VM}$ vs. PBS-T$_{VM}$ ratios at day 7 pvi. Statistical significance calculated using two-way analysis of variance (ANOVA) and Sidak's multiple-comparison test ($^*P < 0.05$, $^{**}P < 0.01$, $^{***}P < 0.001$, $^{****}P < 0.0001$). Data are representative of two to three independent experiments with three to eight mice per group (mean ± s.e.m. in **a-f**, **h**). Each symbol represents an individual mouse; small horizontal lines indicate the mean

cell responses against a subsequent viral[8] or protozoal infection[39]. Nevertheless, these studies did not investigate how helminths could affect bystander memory T cells such as the $T_{VM}$ compartment. Indeed, although $T_{VM}$ development can be highly dependent on IL-4 direct signaling in CD8[+] T cells[22,25], little was known on the ability of this cytokine to drive the expansion of $T_{VM}$ in settings dominated by type 2 immunity such as exposure to helminths. BALB/c mice provide an important opportunity to investigate sensitivity to IL-4 of CD8[+] T cells after egress from the thymus as this strain show a particular sensitivity to IL-4 for driving $T_{VM}$ development[25]. This aspect is of great importance as disregarding IL-4-responsive $T_{VM}$ cells could thwart the full understanding of the mechanism involved and how helminths can affect bystander memory T cells such as $T_{VM}$ cells. Our data reveal that exposure to helminths and derived products elicit Ag-nonspecific expansion of the $T_{VM}$ compartment through direct signaling of IL-4. Notably, we observed that such expansion was associated with Eomes upregulation in $T_{VM}$ cells with no alteration of T-bet expression levels, further characterizing these cells as IL-4-induced $T_{VM}$[17]. Such conditioning was observed in several settings involving Th2-dominating responses such as immunization with helminth Ags or infection with systemic or intestinal helminths like *S. mansoni*, *N. brasiliensis*, or *H. polygyrus*. These observations are important as they demonstrated that helminths could actually condition bystander memory CD8[+] T cells to respond more effectively to their cognate Ags to the benefit of the infected host.

Indeed, our findings revealed that helminths induced Ag-nonspecific expansion of bystander $T_{VM}$ cells in secondary lymphoid organs via direct sensitivity to IL-4 that resulted in increased Ag-specific effector CD8[+] T cell responses in the lungs of BALB/c mice to further enhance the control of MuHV-4 lytic replication. These data are supported by previous reports showing that IL-4Rα deficiency compromises the Ag-specific CD8[+] T cell response to lymphocytic choriomeningitis virus or influenza virus infections whereas the ability of effector IL-4Rα-deficient CD8[+] T cells to kill was unaffected[22,40]. Thus, the inability of *Il4ra*[−/−] mice for early control of MuHV-4 after *S. mansoni* egg exposure lies in the ineffective development of $T_{VM}$ cells and absence of IL-4 responsiveness of CD8[+] T cells in these mice. Although IL-4 signaling in memory T cells was previously associated with downregulation of NKG2D expression and impaired killing capabilities[28], we did not observe significant modulation of NKG2D mRNA expression levels after *S. mansoni* egg treatment in our experimental settings and we observed increased CD8[+] antiviral effector responses in the lung. Our transcriptomics data rather indicated that *S. mansoni* egg-induced $T_{VM}$ were phenotypically distinct from naive $T_{VM}$ cells with increased IFN-γ and IFN-α gene signatures. Nonetheless, their ability to react against their cognate Ag and migrate to the infected lungs was not significantly different to naive $T_{VM}$ in a competitive adoptive transfer in naive congenic mice. These observations suggest that both expansion of the $T_{VM}$ compartment after exposure to helminths and alterations of their phenotype can explain the early response to MuHV-4 Ags and subsequent early control. In addition, the low-proliferation levels in the $T_{VM}$ population after exposure to *S. mansoni* eggs together with the observation that FTY720 did not inhibit $T_{VM}$ expansion in the spleen after IL-4c treatment and $T_{VM}$ expansion in adoptively transferred congenic CD8[+] T lymphocytes suggested that the expansion is likely due to conversion from naive CD44[lo] T cells while independent of proliferation or recruitment to the spleen. As a consequence, expansion of the $T_{VM}$ pool is likely associated with an enrichment of the TCR repertoire in this bystander memory population rather than the proliferation of pre-existing $T_{VM}$ cells, a phenomenon that could significantly augment the probability of

effective Ag priming together with the ability of $T_{VM}$ cells to outcompete their naive CD44[lo] T cell counterpart[30].

Whether helminth-induced bystander $T_{VM}$ cells is also associated with MuHV-4-specific true memory CD8[+] T cells after lytic infection remains unknown, but the establishment of effective memory is rather suggested by a previous report demonstrating that establishment of effective memory against malaria requires IL-4Rα on CD8[+] T cells[41]. This could have important implication for improved vaccination strategies and should further encourage investigators to unravel how $T_{VM}$ cells can be effectively expanded in multiple settings with various types of Ags. Latency during gammaherpesvirus infections such as Kaposi Sarcoma-associated herpesvirus or Epstein–Barr virus, is mainly responsible for malignancies such as lymphomas in human[42,43], a phase during which virus-infected cells evade CD8[+] T cell recognition[44]. However, reactivation events of lytic replication occur and are believed to maintain sufficient levels of infection in the host as well as for transmission[33]. Thus, an effective memory T cell response to control these reactivation events is important. Understanding how helminth-driven expansion of $T_{VM}$ cells impact on long term virus-specific memory is therefore essential.

In conclusion, we provide evidence that helminth-driven type 2 immune responses drives $T_{VM}$ expansion through IL-4 which could in turn positively condition effector Ag-specific CD8[+] T cell responses and significantly enhance the control of viral infection such as lytic gammaherpesvirus infections.

## Methods

**Cells**. Baby Hamster Kidney (BHK)-21 [C13] fibroblasts (ATCC CCL-10) were purchased from ATCC (Manassas, VA, USA) and cultured in Dulbecco's Modified Eagle Medium (DMEM, Sigma-Aldrich) supplemented with 2 mM glutamine, 100 U ml[−1] penicillin, 100 mg ml[−1] streptomycin and 10% fetal calf serum (FCS) at 37 °C under 5% of $CO_2$. Cells were free of mycoplasma contamination (PlasmoTest Mycoplasma Detection Kit, Invivogen).

**Viruses**. The MHV-68 strain of Murid Herpesvirus-4 (MuHV-4) expressing luciferase under the control of the M3 promoter (MuHV-4-Luc)[45] was propagated on BHK-21 cells. Cells and supernatant were harvested at about 4 days post-infection (~85% of lysis) and debris were removed by low-speed centrifugation (1000 × $g$, 10 min, 4 °C). Virions present in the supernatant were harvested by ultracentrifugation (100,000 × $g$, 2 h at 4 °C) and purified through a 30% weight volume[−1] sucrose cushion (100,000 × $g$, 2 h at 4 °C), washed in PBS before being stored in PBS at −80 °C. Viral titers were determined by plaque assay on BHK-21 cells[46]. Briefly, BHK-21 cells monolayers were incubated with 10-fold dilution of viral stock at 37 °C, 5% $CO_2$ for 3 h. Inoculum was then replaced by semisolid medium containing 0.6% carboxymethylcellulose. Cells were further incubated for 4 days then fixed in 4% paraformaldehyde and stained for plaques counting.

To assess antigen-specific CD8[+] T cells responses in BALB/c mice, we modified the MuHV-4-luc virus to insert the H-2K[d]-restricted SYVPSAEQI peptide of the circumsporozoite protein of *P. yoelii* (CSP[280–288]) in frame of the luciferase sequence to generate a MuHV-4-luc-CSP virus strain. First, primers PL451-CSP-STOP-*HindIII*-Fwd (5′-gtcgacggtatcgat*aagctt*agaggaccagggagagcatttgttacaatatcttatgt cccaagcgcggaacaaatataa*aagctt*gatatcgaattccga-3′) and PL451-*HindIII*-STOP-CSP-*HindIII*-Rev (5′-tcggaattcgatatc*aagctt*tatatttgttccgcgcttgggacataagatattgtaacaaatgctctccct ggtcctct*aagctt*atcgataccgtcgac-3′) were annealed and inserted in the HindIII site of the pL451 plasmid vector (http://redrecombineering.ncifcrf.gov/) using inFusion HD kit (Promega) to generate pL451-CSP. Plasmid pL451 contains an FRT-flanked expression cassette of the kanamycin/neomycin resistance gene (KanaR) as selection marker in *Escherichia coli*[47,48]. Then, the CSP sequence was inserted in frame to the carboxy-terminus of the luciferase coding sequence of the MuHV-4-luc BAC clone using an amplicon generated by PCR with primers MuHV-Luc-CSP-Fwd (5′-agatccgcgagattctcattaaggccaagaagggcggcaagatcgccgtgagaggaccagggagagcatt-3′) and MuHV-Luc-CSP-Rev (5′-atgtatctatcatgtctgctcgaagcggccggccgccccgactctagaaatattat gtacctgactgat-3′) and pL451-CSP as template (Supplementary Figure 5h). Phage λ-mediated recombineering in SW105 bacteria (http://redrecombineering.ncifcrf. gov/) followed by arabinose-induced FLP expression for excision of FRT-flanked sequence were used[49]. The generated MuHV-4-luc-CSP BAC construct was verified by an endonuclease restriction and Southern blotting approach and sequencing of the recombination genomic region. The loxP-flanked BAC cassette was removed by virus growth in NIH 3T3-Cre cells to produce MuHV-4-luc-CSP BAC[−] virus[50].

**Parasites**. *S. mansoni*-exposed Swiss-Webster mice and snails were provided by the Schistosome Research Reagent Resource Center for distribution by BEI Resources, NIAID, NIH: *Schistosoma mansoni*, Strain NMRI exposed *Biomphalaria glabrata*, Strain NMRI (NR-21962), *S. mansoni*, Strain NMRI exposed Swiss-Webster mice (NR-21963). *S. mansoni* cercariae were collected from *S. mansoni* exposed *B. glabrata*, and used for natural infection. *S. mansoni* eggs used for egg immunization, were collected from *S. mansoni*-exposed Swiss-Webster mouse liver and stored in PBS at −80 °C, as previously described with minor modifications[27]. Briefly, livers were cut into small pieces with scissors and incubated individually in 50-mL tubes overnight in 20 mL of PBS containing 100 μg ml$^{-1}$ collagenase IV (Sigma-Aldrich). The homogenates were then slowly poured onto strainers of decreasing mesh sizes (425 μm/180 μm/106 μm and 45 μm). Eggs present on top of the 45 μm strainer were collected with PBS and centrifuged for 5 min at 400 × $g$ at RT. Pellets were suspended in 10 mL PBS and layered onto 20% Percoll in 0.25 M sucrose (Sigma-Aldrich) and centrifuged at 800 × $g$ for 10 min at RT. Pellets were then washed in PBS containing 1 mM EGTA and 1 mM EDTA before layered onto 25% Percoll in 0.25 M sucrose and centrifuged at 800 × $g$ for 10 min at RT. Pellets were further washed in PBS, counted and resuspended at 50,000 eggs ml$^{-1}$ and stored in PBS at −80 °C. Soluble Egg Antigen (SEA) from *S. mansoni* were prepared as previously described[51]. Eggs were suspended in PBS at a concentration of 100,000 eggs ml$^{-1}$ and homogenized with a Potter-Elvehjem hand-held homogenizer and centrifuged (2000 × $g$, 20 min at 4 °C). The supernatant was ultra-centrifuged (100,000 × $g$, 90 min at 4 °C) and the final supernatant was filter-sterilized before being stored at −80 °C. Protein concentration was determined by BCA assay (Thermofisher, 23225). Recombinant ω-1 protein was generated in *Nicotiana benthamiana* and purified from the leaf extracellular space using POROS 50 cation resin (Life Technologies)[52].

*N. brasiliensis* was maintained in male Sprague–Dawley rats as described[53]. *N. brasiliensis* L3 larvae were isolated from day 6 to 9 fecal cultures through a Baermann apparatus and used for subcutaneous infection.

*H. polygyrus* was maintained in male CBA mice as described[54]. *H. polygyrus* L3 larvae were isolated from fecal cultures and stored in distilled water at 4 °C.

**Animals**. The experiments, maintenance and care of mice and rats complied with the guidelines of the European Convention for the Protection of Vertebrate Animals used for Experimental and other Scientific Purposes (CETS n° 123). The protocol was approved by the Committee on the Ethics of Animal Experiments of the University of Liège, Belgium (Permit nos. 1357, 1713, and 1849). All efforts were made to minimize suffering. Female BALB/cOlaHsd wild-type mice, 6–8 weeks old and Sprague–Dawley male rats were purchased from Envigo (Venray, Netherlands). Female BALB/B mice (C.B10-*H2$^b$*/LilMcdJ), that are BALB/c congenic for the C57BL/10-derived H-2$^b$ region, were maintained at the Scientific Institute of Public Health, Belgium and transferred to the University of Liege, Department of Infectious Diseases for experiments. BALB/c Cd45.1$^+$ genitor mice were generously provided by Prof. U. Eriksson (Center for Molecular Cardiology, University of Zurich). BALB/c Cd45.1$^+$Cd45.2$^+$ were obtained by crossing Cd45.1$^+$ BALB/c mice with wild-type BALBc/OlaHsd (Cd45.2$^+$). *Il4ra$^{-/-}$* BALB/c mice were bred at the University of Liège, Department of Infectious Diseases. Six- to eight-week-old female littermates were randomly assigned to experimental groups. During experiences, four to five female mice were cohoused per cage, food and water was provided ad libitum. All the animals were bred and/or housed in the University of Liège, Department of Infectious Diseases.

**Mixed bone-marrow chimeric mice models**. Mixed bone-marrow chimeric mice were produced by exposing CD45.1$^+$CD45.2$^+$ BALB/c mice to a whole body lethal irradiation protocol (Gammacell 40 Exactor, 4.5 Gy, two expositions at 3-h interval). The next day, mice were reconstituted by intravenous injection of 4.5 × 10$^6$ BM cells isolated from femurs and tibias of donor CD45.1$^+$ WT and CD45.2$^+$ *Il4ra$^{-/-}$* mice and mixed at a 1:1 ratio. From 1 week before to 3 weeks after irradiation, mice were given broad-spectrum antibiotherapy (0.27 g trimethoprimum and 1.33 g sulfa-diazinum per liter of drinking water, Emdotrim 10%, Ecuphar). Mice were left untreated for 8 weeks to allow complete reconstitution and chimerism of the different blood leukocytes populations was then confirmed by flow cytometry.

**Helminth infections and immunization with helminth Ags**. For natural infection with *S. mansoni*, mice were anesthetized by intraperitoneal injection of 16 mg kg$^{-1}$ of xylazine and 100 mg kg$^{-1}$ of ketamine and infected by percutaneous exposure to 35 cercariae during 30 min. Treatment with *S. mansoni* eggs consisted of an intraperitoneal immunization on day 0 (5000 eggs per mouse) followed by one intravenous injection of 5000 eggs on day 14. In some experiments, mice received *S. mansoni* eggs (5000 eggs per mouse), SEA (60 μg per mouse) or recombinant ω-1 (10 μg per mouse) intraperitoneally at 2 weeks interval.

Mice anesthetized by isoflurane inhalation were infected with *N. brasiliensis* by subcutaneous injection of 500 × L3 larvae. In some experiments, mice were treated with ivermectin (10 mg L$^{-1}$ of drinking water, Noromectin, Norbrook) from day 14 to 21 after *N. brasiliensis* infection. Mice were infected with *H. polygyrus* by oral gavage (200 × L3).

**IL-4 complex treatment**. Mice received 2 intraperitoneal injections of IL-4c: 5 μg of recombinant IL-4 (BioLegend, carrier-free, Cat # 574306) and 25 μg of anti-IL-4 antibody (BioLegend, clone 11B11, LEAF purified, Cat # 504108) per mouse at days 0 and 2.

**FTY720 treatment**. In order to block lymphocyte trafficking, mice received daily i. p. administration of 1 mg kg$^{-1}$ of FTY720 (Sigma-Aldrich) dissolved in 100 ml of sterile water as described elsewhere[55].

**Peptide-MHC class I tetramer enrichment**. Enrichment of Ag-specific CD8$^+$ T cells was performed using an established protocol[22]. Combined spleen and lymph nodes were digested with collagenase D. Cells were then labeled with APC-conjugated H-2K$^d$-CSP$^{280-288}$ tetramer (SYVPSAEQI, NIH Tetramer Core Facility) for 30 min at room temperature, followed by magnetic enrichment over LS columns using anti-APC microbeads (Miltenyi biotec). Enriched samples and unbound fractions were stained with surface antibodies, and Polybead polystyrene microspheres (Polyscience) were used for calculating cell number.

**Viral infection and quantification**. Mice were anesthetized with isoflurane and 10$^4$ PFU (10$^3$ or 10$^5$ PFU for some experiments) of MuHV-4-Luc or MuHV-4-Luc-CSP was administered intranasally in 30 μl of PBS. Light emission was then monitored by in vivo bioluminescence imaging using a IVIS Spectrum In Vivo Imaging System (Perkin Elmer) after D-luciferin intraperitoneal injection (75 mg kg$^{-1}$, Perkin Elmer). Living Image v4.1 software (Perkin Elmer) was used to obtain the total flux (photons s$^{-1}$) using a fixed-sized region of interest or average radiance (photons s$^{-1}$ cm$^{-2}$ steradiants$^{-1}$).

Infectious virus in lungs was quantified after homogenization using glass Potter-Elvehjem hand-held homogenizers after freezing (−80 °C) in 6 ml complete medium prior to plaque assay[46].

Viral genome loads were measured using a Taqman-based real-time PCR reaction (CFX)[56]. DNA was extracted from spleens using Wizard genomic DNA purification kit (Promega) and 100 ng per sample was used in iQ supermix (Biorad) to amplify a MuHV-4-specific sequence (genomic co-ordinates 43,015–43,138, NC_001826.2): gene ORF25 with primers ORF25fwd, 5′-atggtatagccgcctttgtg-3′ and ORF25rev 5′-acaagtggatgaagggttgc-3′ and probe 5′-6-carboxyfluorescein [FAM]-caaccactggatcagcataaaacttatgaa-black hole quencher [BHQ1]-3′. Cellular DNA was quantified in parallel by amplifying the interstitial retinoid binding protein (IRBP) gene sequence using iQ SYBR Green supermix (Biorad) and IRBPfwd 5′-atccctatgtcatctcctacytg-3′ and IRBPrev 5′-ccrctgccttcccatgtytg-3′ primers. Standard curves were obtained using pGEMT-easy (Promega) plasmid templates in which MuHV-4 ORF25 or mouse IRBP sequences were cloned, respectively. Amplified products were distinguished from paired primers by melting curve analysis and the correct sizes of the amplified products confirmed by electrophoresis and staining with ethidium bromide.

**Histology and immunohistochemistry**. Lungs were collected from PBS-perfused animals and immediately fixed in 10% neutral buffered formalin. Tissues were embedded and sectioned, and sections were stained with hematoxylin and eosin. Immunochemistry was performed using EnVision Detection Systems (DAKO) with anti-MHV-68 rabbit polyserum or naive rabbit serum as primary antibodies.

**Antigen-specific antibody titers**. Nunc Maxisorp ELISA plates (Nalgene Nunc) were coated overnight at 4 °C with MuHV-4 virions (10$^6$ PFU ml$^{-1}$ of carbonate buffer pH 9.5 containing 0.1% Triton X-100), or SEA (10 μg mL$^{-1}$ in carbonate buffer) before being incubated for 1 h in wash/blocking buffer (0.1% Tween-20 and 3% BSA in PBS) at RT. Plates were then incubated with mouse sera (serial dilutions) in wash/blocking buffer for 2 h at RT. Detection was performed using alkaline phosphatase-conjugated anti-mouse IgG1 or IgG2a (BD Biosciences) in wash/blocking buffer for 1 h at RT. Chromogenic reaction was performed using p-Nitrophenylphosphate (Sigma) and stopped in NaOH 1 M before absorbance was read at 405 nm using a iMark ELISA plate reader (Biorad).

**BAL cytokines quantification**. After euthanasia, airways were flushed twice with 1 ml of ice cold PBS containing protease inhibitors (cOmplete, Roche) via catheterization of the trachea. Quantification of IL-4, IFN-γ and granzyme B was performed using specific Ready-SET-Go kits (eBioscience, Cat # 88-8022-88, Cat # 88-7314-88, and Cat # 88-7044-88) following the manufacturer's instructions.

**Tissue processing and cell preparation**. Airways were flushed twice with 1 ml of ice cold PBS, cells were harvested by centrifugation of the BAL fluid. After section of the vena cava, lungs were perfused with 5 ml of ice-cold PBS through the right ventricle. Lung were dissociated with the gentleMACS dissociator (Miltenyi Biotec) in C-tubes (Miltenyi Biotec), incubated in HBSS (Gibco), 5% FCS, 1 mg ml$^{-1}$ of collagenase D (Roche), and 0.1 mg ml$^{-1}$ of DNase I (Roche) for 30 min at 37 °C under agitation and further dissociated with the gentleMACS dissociator. The resulting suspension was washed in cold PBS with 2 mM EDTA and filtered on a 100 μm cell strainer (Falcon). Spleen and cranial and caudal mediastinal lymph nodes were disrupted using scissors and a sterile syringe plunger and filtered

through a 70 μm filter. Erythrocytes were lysed in red cell lysis solution (155 mM NH$_4$CL, 0.12 mM EDTA, 10 mM KHCO$_3$) and counted in a Neubauer cytometer chamber with trypan blue 0.4% dye for exclusion of dead cells.

**Flow cytometry and cell sorting**. Incubations were performed in FACS buffer (PBS containing 0.5% BSA and 0.1% NaN$_3$) at 4 °C. Cells were first incubated with anti-mouse CD16/32 antibody (clone 93, 1 μg ml$^{-1}$, BioLegend, Cat # 101302) before fluorochrome-conjugated antibodies against surface antigens were added and incubated during 20 min at 4 °C. Various panels were used including antibodies to CD3e (clone 145-2C11, 0.4 μg ml$^{-1}$, APC, Cat# 17-01031-81), CD19 (MB19-1, 0.4 μg ml$^{-1}$, APC, Cat # 17-0191-82), CD4 (RM4-5, 0.4 μg ml$^{-1}$, PerCP-Cy5.5, Cat # 45-0042-82), CD44 (IM7, 0.4 μg ml$^{-1}$, PE, Cat # 12-0441-82), CD49d (R1-2, 2.5 μg ml$^{-1}$, biotinylated, Cat # 13-0492), CD62L (MEL-14, 0.8 μg ml$^{-1}$, eFluor 450, Cat # 48-0621-82), MHC class II (M5/114.15.2, 0.1 μg ml$^{-1}$, PE-Cy7, Cat # 25-5321-82 or eFluor 450, Cat # 48-5321-82), Gr-1 (RB6-8C5, 1 μg ml$^{-1}$, FITC, Cat # 11-5931-82), CD11c (N418, 0.8 μg ml$^{-1}$, PerCP-Cy5.5, Cat # 45-0114-80 or Alexa Fluor 700, Cat # 56-0114-82), Ly6C (HK1.4, 0.2 μg ml$^{-1}$, PE-Cy7, Cat # 25-5932-80), CD11b (M1/70, 0.4 μg ml$^{-1}$, APC-eFluor 780, Cat # 47-0112-80) all from eBioscience/ThermoFisher Scientific; antibodies to CD3 molecular complex (17A2, 0.4 μg ml$^{-1}$, v450, Cat # 561389), CD19 (1D3, 0.4 μg ml$^{-1}$, APC-Cy7, Cat # 557655), CD8a (53-6.7, 1 μg ml$^{-1}$, FITC, Cat # 553031), CD183 (CXCR3-173, 2 μg ml$^{-1}$, PE, Cat # 562152), KLRG1 (2F1, 0.4 μg ml$^{-1}$, PE, Cat # 561621), CD127 (SB/199, 4 μg ml$^{-1}$, BV786, Cat # 563748), Siglec-F (E50-2440, 0.4 μg ml$^{-1}$, PE, Cat # 552126 or PE-CF594, Cat # 562757), CD45.1 (A20, 2 μg ml$^{-1}$, APC, Cat # 558701), CD45.2 (104, 0.4 μg ml$^{-1}$, v500, Cat # 562129) all from BD Biosciences and antibody to CD44 (IM7, 0.4 μg ml$^{-1}$, PE-Cy7, Cat # 103030), and CD45.1 (A20, 0.8 μg ml$^{-1}$, BV421, Cat # 110732) from BioLegend. Biotinylated antibodies were detected using BV650-conjugated streptavidin (0.2 μg ml$^{-1}$, BD Biosciences, Cat # 563855). Dead cells were stained using Fixable Viability Stain 510 (0.4 μg ml$^{-1}$, BD Bioscience, Cat # 564406) or Fixable Viability Dye eFluor 780 (1000× dilution, eBioscience, Cat # 65-0865-14). In experiments in which intranuclear staining for transcription factors was needed, cells were fixed and permeabilized using Foxp3/ Transcription factor staining buffer set (eBioscience, Cat # 00-5523-00) following manufatuer's instruction and incubated 30 min at 4 °C with antibody against either T-bet (O4-46, 100× dilution, PE, BD Biosciences, Cat # 561268) or Eomes (Dan11mag, 2 μg ml$^{-1}$, PE, eBioscience, Cat # 12-4875-82) diluted in permeabilization buffer. Samples were analyzed on a BD LSR Fortessa X-20 flow cytometer, cell sorting was performed on a FACS Aria IIIu (BD biosciences).

**In vivo EdU incorporation**. To assess the proliferation of virtual memory CD8$^+$ T lymphocytes upon helminth exposure, *S. mansoni* eggs exposed mice were injected ip with EdU (500 μg per mouse in PBS, ThermoFisher Scientific) 4 h before endpoint (1×) or daily for 4 days before endpoint (4×). Mice were euthanized and spleen were processed as described above. Surface staining was performed with exception of PE-conjugated antibodies. After dead cells staining, cells were fixed for 15 min at room temperature in Click-It Fixative (ThermoFisher Scientific), washed with PBS/1%BSA and incubated 15 min at RT in Click-It Saponine-Based Permeabilization Buffer (ThermoFisher Scientific). Cells were then resuspended in freshly prepared EdU staining cocktail [10 μM Pacific Blue-azide, 1 mM CuSO$_4$, 10 mM sodium ascorbate with 1 mM THPTA (tris((1-hydroxy-propyl-1H-1,2,3-triazol-4-yl)methyl)amine) and 10 mM amino-guanidine in PBS] and incubated 2 h at RT[57]. Cells were washed with Click-It Saponine-Based Permeabilization Buffer and incubated with PE-conjugated antibodies for 20 min at 4 °C. Samples were analyzed on a BD LSR Fortessa X-20 flow cytometer (BD Biosciences).

**MHC-tetramer stainings**. Lung and BAL cells were processed as described above from BALB/B mice (H-2$^b$ congenic BALB/c). Cells were incubated with BV421-conjugated tetramers H-2D$^b$-ORF6$^{487-495}$ (AGPHNDMEI, 90 nM) or H-2K$^b$-ORF61$^{521-531}$ (TSINFVKI, 45 nM), or APC-conjugated H-2K$^d$-CSP$^{280-288}$ (80 nM) (NIH Tetramer Core Facility), respectively, for 30 min at RT before further staining. Samples were analyzed on a BD LSR Fortessa X-20 flow cytometer (BD Biosciences).

**Ex vivo restimulation and cytokine production**. Cytokine production upon restimulation was assessed by intracellular cytokine staining (ICCS) and flow cytometry or by ELISA on culture supernatants. For ICCS, cells were cultured at 37 °C, 5% CO$_2$ in IMDM complemented with 2 mM glutamine, 100 U penicillin ml$^{-1}$, 100 mg streptomycin ml$^{-1}$ and 10% FCS for 4 h in presence of brefeldin A (10 μg mL$^{-1}$, eBioscience, Cat # 00-4506-51), monensin (2 μM, eBioscience, Cat # 00-4505-51), and restimulating agent. For unbiased restimulation, cells were incubated with phorbol 12-myristate 13-acetate (PMA, 20 ng mL$^{-1}$, Sigma-Adlrich, Cat # P8139) and ionomycin (1 μg mL$^{-1}$, Sigma-Adlrich, Cat # I0634). For antigen-specific restimulation, cells were incubated with H-2$^b$-restricted MuHV-4 ORF6$^{487-495}$ (AGPHNDMEI) and ORF61$^{521-531}$ (TSINFVKI) peptides (1 μM). Following surface and viability stainings, cells were fixed in 2% paraformaldehyde overnight and washed with Permeabilization Buffer (eBioscience) before being incubated with antibodies against IFN-γ (clone XMG1.2, 2 μg ml$^{-1}$, PE, BioLegend, Cat # 505808) and TNF-α (clone MP6-XT22, 0.67 μg ml$^{-1}$, BV711, BD Biosciences, Cat # 563944) in Permeabilization Buffer for 20 min at 4 °C. For ELISA, cells were

cultured at 37 °C, 5% CO$_2$ in IMDM complemented with 2 mM glutamine, 100 U penicillin ml$^{-1}$, 100 mg streptomycin ml$^{-1}$ and 10% FCS for 72 h with SEA (20 μg mL$^{-1}$). Supernatant were collected and conserved at −20 °C. Quantification of cytokines IL-4 and IFN-γ was performed using specific ELISA (Ready-SET-Go, eBioscience) following manufacturer's instructions.

**In vivo CD8$^+$ or CD4$^+$ cells depletion**. For depletion experiments, mice were injected intraperitoneally with anti-CD8 (clone YTS-169.4, 200 μg per injection) or anti-CD4 (clone GK1.5, 200 μg per injection) both obtained from BioXCell at day −1, 1 and 4 after MuHV-4-Luc infection. Depletion was confirmed by flow cytometry analysis of blood cells on day 6.

**T cell enrichment and competitive adoptive transfer**. Spleen single-cell suspensions were obtained by forcing through a 70 μm cell strainer (Falcon) before cell counts determined. CD8$^+$ T cells were then enriched using negative magnetic isolation (Miltenyi Biotec) and used either for adoptive transfer of bulk CD8$^+$ T cells or for further T$_{VM}$ purification: enriched CD8$^+$ T cells were then FACsorted to isolate CD44$^{hi}$CXCR3$^{hi}$CD62L$^{hi}$CD49d$^{lo}$ T$_{VM}$ cells to high purity (>97%) before further analysis or adoptive transfer. Competitive adoptive transfer of naive and Sm egg-induced T$_{VM}$ cells consisted in the intravenous administration into WT naive BALB/c congenic recipients of equal numbers ($4 \times 10^6$ cells total) of naive and Sm egg-induced T$_{VM}$ cells. For competitive transfer of WT and *Il4ra*$^{-/-}$ bulk CD8$^+$ T cells, equal number ($4 \times 10^6$ cells total) of WT and *Il4ra*$^{-/-}$ enriched CD8$^+$ T cells from naïve or Sm egg-treated BALB/c mice were intravenously administered respectively into naive or Sm egg-treated WT BALB/c congenic recipients.

**RNA sequencing**. T$_{VM}$ cells were extracted as described in the previous section before RNA extraction with on-column DNase treatment (RNeasy Plus mini kit, Qiagen). Integrity of extracted RNAs was controlled on a 2100 Agilent Bioanalyzer and samples with a RNA integrity number > 9 were processed for RNA sequencing. Libraries were prepared from 0.5 μg of RNA of three independent replicates per group, using the Illumina TruSeq Stranded mRNA library preparation kit. Libraries were then sequenced and bioinformatics analysis was performed. Approximately 30 million 75-base single-end reads were generated per sample. Quality control checks on raw sequencing data for each sample were performed using FastQC (Babraham Bioinformatics). Reads were mapped to the mouse reference genome (mm10) using STAR (version 3.4.0)[58]. Subsequently the analysis was performed with R Bioconductor packages:[59] Rsamtools (version 1.18.3) and GenomicAlignments (version 1.2.2) were used to count the reads by exons, and gene count datasets were then analyzed to determine DE genes (DEGs) using DESeq2 (version 1.16.1). A gene was determined to be a DEG by passing FDR < 0.1 and log 2-fold change ≥ ±0.5.

**Statistical analysis**. Statistical evaluation of different groups was performed either by analysis of variance (ANOVA) followed by the Dunnett or Sidak multiple-comparison test or by nonparametric Mann–Whitney test, as indicated. A *P* value < 0.05 was considered significant. Statistical analyses were performed using Prism v6 (Graphpad, La Jolla, CA).

## Data availability
Sequence data that support the findings of this study have been deposited in the Gene Expression Omnibus (GEO) repository with the primary GSE110971 accession code. All other data are available from the authors upon request.

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

## Acknowledgments

M.R. is a Research Fellow of the Fonds pour la formation à la Recherche dans l'Industrie et dans l'Agriculture (FRIA). A.M.D. and T.C. were supported by a BeIPD ULiege-Marie Curie COFUND fellowship. B.D. is a Research Associate of the Fonds de la Recherche Scientifique (F.R.S.-FNRS). The authors are thankful to Lorène Dams, Cédric Delforge, Aurélie Vanderlinden, and Emeline Deglaire for technical assistance. The following tetramers were obtained through the NIH Tetramer Facility: H-2D$^b$-ORF6$^{487–495}$, H-2K$^b$-ORF61$^{524–531}$, and H-2K$^d$-CSP$^{280–288}$. *B. glabrata* snails were provided by the NIAID Schistosomiasis Resource Center of the Biomedical Research Institute (Rockville,

MD) through NIH-NIAID Contract HHSN272201000005I for distribution through BEI Resources. R.M.M. is supported by a Wellcome Trust investigator award (Ref. 106122), and Wellcome Trust core funding to the Wellcome Centre for Molecular Parasitology (Ref. 104111).

## Author contributions

M.R. and A.M.D. designed and performed the experiments, analyzed the data, prepared the figures, and wrote the paper. A.C. and T.C. designed and performed experiments and analyzed the data. B.M. performed experiments and contributed to manuscript preparation. J.J. supervised the parasite lifecycles and performed experiments. M.S. and R.M. provided support for parasite lifecycles and contributed to manuscript preparation. X.X. provided support for the analysis of RNA sequencing data. C.H. provided the schistosome omega-1 recombinant proteins. O.D. supervised mouse breeding and provided the C.B10-$H2^b$/LilMcdJ BALB/B mice. F.B. supervised mouse breeding and provided $Il4ra^{-/-}$ mice and contributed to manuscript preparation. A.V., L.G., and W.G.H. contributed the design of experiments and to manuscript preparation. B.G.D. planned and supervised the work, acquired funding, designed experiments, and wrote the paper.

## Additional information

**Competing interests:** The authors declare no competing interests

