## [Peer Review File · Nature Communications]

REVIEWERS' COMMENTS:

Reviewer #1 (Remarks to the Author):

I am compelled by the authors rebuttal and the additional data they have included in the manuscript! I admit i had not considered the novelty associated with the fact that a Th2 centric parasitic infection could advantageously condition a host for enhanced host protection against viral challenge. Indeed, I would encourage the authors to punch that point a bit more aggressively since their formal demonstration of this really holds the majority (in my view) of their novelty and impact. for example, the last paragraph of the discussion i suggest would make a much better first paragraph of their discussion or even the final paragraph of their introduction... this would give the reader the right perspective from which they want the importance their data assessed. While i would still prefer that some better mechanistic data were included (eg.IL-12R-/- or IL-18R-/- host to verify if Tvm bystander IFNg is critical? NKG2D blockade to determine if bystander lytic function is required) i do like the additions of the Edu and FTY data in affirming that this is a Tvm conversion scenario, not a Tvm proliferation, a point they might also push a bit more in the discussion as it further validates the previous conclusions from Hildeman and colleagues (tripathi et al. EJI 2016) that IL-4 was more important in the Balbc host for Tvm differentiation/conversion than IL-15. another point worth making is the fact that the BalbC results seems a bit different than the B6 in terms of what T cell subset Tvms are more likely to produce after activation... the authors here show that the egg-injected hosts have more SLEC which contrasts with B6 mice in which Tvm are more likely to form Tcm (Lee et al. PNAS). these are all just suggestions of course, but i think they would help highlight the novelty as well as integrate the data into the existing literature on Tvms.

lastly, a minor point but perhaps important, at least to its authors... the reference Haluszczak et al. JEM 2009 is when Tvms were first identified and the term "Virtual Memory" was coined. As such, it should probably be added to the introduction where Tvms are first discussed.

collectively, my complements to the authors on a well done rebuttal and revision.

Reviewer #2 (Remarks to the Author):

I had previously reviewed this manuscript for Nature Immunology and found the findings to be very interesting. My opinion has not changed.

In the manuscript ., the authors describe a fascinating phenomenon whereby helminth infection or exposure to helminth antigens, via IL-4 stimulation, will non-specifically expand a population of CD8+ T cells, previously described as virtual memory T cells. This results in enhanced protection against viral infections. This finding run contrary to other reports whereby helminth infection have a detrimental effect on anti-viral responses. In this manuscript, the opposite is observed, hence there is much complexity to the effects of co-infection that might well also be mouse strain dependent. Using the tetramers to an irrelevant epitope is a nice touch. I do wonder about other tetramers (e.g. LCMV) but this is not necessary. The data supporting the main findings in this manuscript is mostly quite strong, certainly, the requirement for IL-4 in this process is elegantly demonstrated through a variety of gain of function and loss of function experiments culminating with a set of mixed transfer experiments. This could have been further improved with CD8 specific deletion of the IL-4R using a floxed allele, but the data as it is, is quite convincing. Overall, this manuscript adds substantially to our understanding of co-infection, at least in the sense that it opens up the discussion as to the complexity of whether co-infections can be good or bad, and provide a mechanism for explaining enhanced anti-viral responses during helminth infection, which may certainly be very dependent on the context of the co-infection.

There are nonetheless some concerns that could be improved on the manuscript as described below:

1) It is not clear to me the kinetics of this relationship, perhaps due to the use of the term virtual

memory. The author has provided some additional information but more might be needed. How long does the enhanced viral immunity last? How long do the non-specifically expanded CD8+ T cells persist for? Does the non-specifically increased tetramer positive T cells decline after the inevitable loss/decline of the Type 2 response? This seems to be a major unanswered question. This issue is somewhat addressed in the experiments shown in Figure S3, but there was no quantification of the CD8 Tvm population in those experiments.

2) The other puzzling issue is the low proliferation of the Tvm population despite the significant expansion with S.m. eggs. The kinetics of where they expand could be addressed further by timing the Edu or BrDu incorporation (longer) to see if they are expanding somewhere else. Since the authors have shown that they can rapidly expand the CD8 Tvm population just by giving IL-4c, this should be quite straight forward to examine where the expansion takes place, or it is a trafficking issue.

Point-by-point response to the reviewers' comments

Reviewer #1 (Remarks to the Author):

I am compelled by the authors rebuttal and the additional data they have included in the manuscript! I admit i had not considered the novelty associated with the fact that a Th2 centric parasitic infection could advantageously condition a host for enhanced host protection against viral challenge. Indeed, I would encourage the authors to punch that point a bit more aggressively since their formal demonstration of this really holds the majority (in my view) of their novelty and impact. For example, the last paragraph of the discussion I suggest would make a much better first paragraph of their discussion or even the final paragraph of their introduction... this would give the reader the right perspective from which they want the importance their data assessed. While I would still prefer that some better mechanistic data were included (eg.IL-12R-/- or IL-18R-/- host to verify if Tvm bystander IFNg is critical? NKG2D blockade to determine if bystander lytic function is required) I do like the additions of the Edu and FTY data in affirming that this is a Tvm conversion scenario, not a Tvm proliferation, a point they might also push a bit more in the discussion as it further validates the previous conclusions from Hildeman and colleagues (tripathi et al. EJI 2016) that IL-4 was more important in the Balbc host for Tvm differentiation/conversion than IL-15.

Another point worth making is the fact that the BalbC results seems a bit different than the B6 in terms of what T cell subset Tvms are more likely to produce after activation... the authors here show that the egg-injected hosts have more SLEC which contrasts with B6 mice in which Tvm are more likely to form Tcm (Lee et al. PNAS). these are all just suggestions of course, but i think they would help highlight the novelty as well as integrate the data into the existing literature on Tvms.

Lastly, a minor point but perhaps important, at least to its authors... the reference Haluszczak et al. JEM 2009 is when Tvms were first identified and the term "Virtual Memory" was coined. As such, it should probably be added to the introduction where Tvms are first discussed.

Collectively, my complements to the authors on a well done rebuttal and revision

Authors response. We thank the reviewer for the positive response given to our revised manuscript and rebuttal. We appreciate the suggestions to move the last paragraph of our discussion to the beginning of the discussion instead and we have revised our manuscript accordingly. We also thank the reviewer for pointing out the reference Haluszczak et al. JEM 2009 that we now cite in the introduction when mentioning Tvms.

Regarding the suggestion to emphasize the difference between BALB/c and C57BL/6 mice on the ability to form short-lived memory T cells or central memory T cells, we find it highly interesting. However, we prefer to leave this discussion out as we believe a number of important questions can be raised when thinking of the differences between C57BL/6 and BALB/c and the actual suggestion by the reviewer is at this stage speculative and we did not address this directly.

Finally, we did cite the work of David Hildeman (Tripathi et al. , EJI 2016) for the importance of IL-4 in Balb/c mice for Tvm differentiation.

Reviewer #2 (Remarks to the Author):

I had previously reviewed this manuscript for Nature Immunology and found the findings to be very interesting. My opinion has not changed.

Authors response. We thank the reviewer who is including below the comments that were made after the initial review. We have addressed all the comments that are detailed in our previous point-by-point response. As we can read from the interest given to our work, we believe this reviewer is happy with the revisions.

In the manuscript, the authors describe a fascinating phenomenon whereby helminth infection or exposure to helminth antigens, via IL-4 stimulation, will non-specifically expand a population of CD8+ T cells, previously described as virtual memory T cells. This results in enhanced protection against viral infections. This finding runs contrary to other reports whereby helminth infection has a detrimental effect on anti-viral responses. In this manuscript, the opposite is observed, hence there is much complexity to the effects of co-infection that might well also be mouse strain dependent. Using the tetramers to an irrelevant epitope is a nice touch. I do wonder about other tetramers (e.g. LCMV) but this is not necessary. The data supporting the main findings in this manuscript is mostly quite strong, certainly, the requirement for IL-4 in this process is elegantly demonstrated through a variety of gain of function and loss of function experiments culminating with a set of mixed transfer experiments. This could have been further improved with CD8 specific deletion of the IL-4R using a floxed allele, but the data as it is, is quite convincing. Overall, this manuscript adds substantially to our understanding of co-infection, at least in the sense that it opens up the discussion as to the complexity of whether co-infections can be good or bad, and provide a mechanism for explaining enhanced anti-viral responses during helminth infection, which may certainly be very dependent on the context of the co-infection.

There are nonetheless some concerns that could be improved on the manuscript as described below:

1) It is not clear to me the kinetics of this relationship, perhaps due to the use of the term virtual memory. The author has provided some additional information but more might be needed. How long does the enhanced viral immunity last? How long do the non-specifically expanded CD8+ T cells persist for? Does the non-specifically increased tetramer positive T cells decline after the inevitable loss/decline of the Type 2 response? This seems to be a major unanswered question. This issue is somewhat addressed in the experiments shown in Figure S3, but there was no quantification of the CD8 Tvm population in those experiments.

2) The other puzzling issue is the low proliferation of the Tvm population despite the significant expansion with S.m. eggs. The kinetics of where they expand could be addressed further by timing the Edu or BrDu incorporation (longer) to see if they are expanding somewhere else. Since the authors have shown that they can rapidly expand the CD8 Tvm population just by giving IL-4c, this should be quite straightforward to examine where the expansion takes place, or it is a trafficking issue.